# COVID-19 aerosol transmission simulation-based risk analysis for in-person learning

**Tessa Swanson**[1]*, **Seth Guikema**[1,2], **James Bagian**[1,3], **Christopher Schemanske**[1], **Claire Payne**[1]

**1** Industrial and Operations Engineering, University of Michigan, Ann Arbor, Michigan, United States of America, **2** Civil and Environmental Engineering, University of Michigan, Ann Arbor, Michigan, United States of America, **3** Anesthesiology, University of Michigan, Ann Arbor, Michigan, United States of America

* tlswan@umich.edu

**Data Availability Statement:** The methods and source code used to produce the results and analyses presented in this paper with artificial data are available on GitHub at: https://github.com/tlswan/in-class_covid_transmission. Data cannot

## Abstract

As educational institutions begin a school year following a year and a half of disruption from the COVID-19 pandemic, risk analysis can help to support decision-making for resuming in-person instructional operation by providing estimates of the relative risk reduction due to different interventions. In particular, a simulation-based risk analysis approach enables scenario evaluation and comparison to guide decision making and action prioritization under uncertainty. We develop a simulation model to characterize the risks and uncertainties associated with infections resulting from aerosol exposure in in-person classes. We demonstrate this approach by applying it to model a semester of courses in a real college with approximately 11,000 students embedded within a larger university. To have practical impact, risk cannot focus on only infections as the end point of interest, we estimate the risks of infection, hospitalizations, and deaths of students and faculty in the college. We incorporate uncertainties in disease transmission, the impact of policies such as masking and facility interventions, and variables outside of the college's control such as population-level disease and immunity prevalence. We show in our example application that universal use of masks that block 40% of aerosols and the installation of near-ceiling, fan-mounted UVC systems both have the potential to lead to substantial risk reductions and that these effects can be modeled at the individual room level. These results exemplify how such simulation-based risk analysis can inform decision making and prioritization under great uncertainty.

## Introduction

Educational institutions such as universities and K-12 schools face ongoing challenges in managing risks associated with the ongoing SARS-CoV-2 pandemic. As schools and universities return to in-person classes they must decide what, if any, mitigation measures to implement. Should they require masks? Reduce density in classes? Install in-room active mitigation measures such as HEPA filtration or UVC virus inactivation? What testing policy should they have? These types of decisions may have significant impacts on virus spread, institutional operations, and cost, yet they must be made under localized conditions of considerable uncertainty

be shared publicly because the minimal underlying data contain potentially re-identifiable sensitive information and is regulated by IRB under ID HUM00206822. Data is available upon request through the open platform of the Inter-university Consortium for Political and Social Research (openICPSR) under ID openicpsr-172081. Requests for data access can be sent through https://www.openicpsr.org/openicpsr/project/172081/version/V2/view.

**Funding:** This work is funded by the University of Michigan College of Engineering's COVID-19 Skunkworks project, which provided the data for analysis but does not hold any role the preparation or publishing of this manuscript.

**Competing interests:** The authors have declared that no competing interests exist.

including temporal variability about virus spread, prevalence of the virus in the surrounding community, and the health impacts of infections.

Risk analysis can provide valuable support for decision-makers by providing both quantitative and qualitative insights into the risks faced by decision makers and the impacts of different mitigation measures. Here we use the term "risk" in line with the accepted definitions from the Society for Risk Analysis [1]. Risk must consider both the potential outcome space (what can happen), the uncertainty about the outcomes, and the severity of the outcomes (how much do we care about them) in a given situation. A key question then is what the end points of concern are. In this work we consider the key outcomes to be health effects of infections with infections themselves being an important intermediate measure. That is, we cannot use only infections as the end point of the analysis; we must consider health outcomes such as hospitalization and death. At the same time, we do not explicitly consider other critical outcomes such as student learning and mental health outcomes and institutional financial and reputation outcomes among many others. These are all critical components of the landscape of risk associated with in-person teaching during COVID-19 and must be considered in addition to the output of quantitative risk models like the one presented here for the ultimate decision and policy making.

One aspect of many risk analyses is a quantitative risk model. While a quantitative risk model does not address all aspects of a given situation, it can help by providing probabilistic estimates of outcomes such as, in the case of COVID, infections, hospitalizations, and deaths due to virus aerosol transmission under different mitigation strategies. Doing so requires that the risk analysis approach models transmission at the level of individuals within classrooms to allow the room-level effects of interventions and mixing of individuals across classes to be modeled. Also, influences such as the prevalence of infection in the outside community must be accounted for. In addition, the risk analysis approach must provide estimates of the uncertainty in the assessed risks in a way that reflects the high degree of uncertainty in virus transmission. This requires a fundamentally different approach than traditional compartmentalized, population-averaged epidemiology-based models.

In this paper we develop a stochastic simulation framework to estimate the probability of infection, hospitalization, and death at the individual level and collective (campus) levels and to compare these risks with different potential mitigation measures. This framework simulates transmission between individual students and faculty in each meeting of each class throughout a semester, resulting in estimated probability distributions of both individual likelihood of becoming infected, hospitalized, or dying and estimated probability distributions for these three outcomes for the campus as a whole. It must be emphasized that these estimated probabilities are highly dependent on the input values for parameters such as community prevalence of disease, transmissibility of the disease vector, and testing policy, among others. This approach provides support that informs both individuals as well as those responsible for higher level policy decisions as they choose among possible individual-level and room-level and organization-level mitigation options.

We demonstrate our approach by modeling COVID risk for one semester of operation of classes in a real college with approximately 11,000 students set in the context of a larger university. For privacy reasons we leave this college unnamed given the sensitive nature of this work. While this model was initially designed to support mitigation decision making related to the SARS-Cov-2 virus in a college setting, it can be used to model any other aerosol-transmitted virus as long as the needed parameter values can be satisfactorily approximated. It can also be used in other settings such as primary and secondary schools with appropriate adjustments in model logic and input parameters. This model advances the ability to estimate risks associated with virus transmission in a way that allows mitigation measures at the individual and room

levels to be evaluated and compared, providing stronger support for mitigation decision making as well as communicating the underlying rationale to the myriad stakeholders to facilitate ultimate organizational success.

An important caveat is needed up front. No model can fully represent reality. Or in the words of a well-known quote often attributed to George Box "All models are wrong, but some are useful" [2]. The goal of a quantitative risk model such as the one presented here is not to make precise predictions about future outcomes. Rather, the purpose is to provide a probabilistic understanding of potential outcomes under different interventions and, most critically, to provide a relative ranking of risk and better understanding of the effects of different potential interventions thus aiding in the selection of the most prudent decisions.

This paper is organized as follows. We first present background on COVID-19 aerosol transmission, dose response modeling, and simulating COVID-19 transmission and response at universities. We next present the simulation model and the data used to populate this model. We then demonstrate the model using the the course schedule for an actual semester of classes in our example college. Following this, we discuss the results and the limitations in the current version of the model. We conclude with comments on the benefits of our model and simulation-based risk analysis in general and identify opportunities for expanding this methodology to other settings. We provide additional details about the model's algorithmic structure, input parameters, and detailed model results for the example college in the Appendices.

## Background

Consensus has built around the dominance of SARS-CoV-2 transmission via aerosols [3, 4]. In addition, while symptoms such as coughing and sneezing lead to higher levels of virus being expelled by those who are infected, the high rate of asymptomatic prevalence of SARS-CoV-2 requires consideration of spread beyond quarantining those who express symptoms [5, 6]. Just one seemingly healthy but infected individual is capable of transmitting SARS-CoV-2 through breathing and that risk increases with higher respiratory activity from whispering, to lecturing, to singing [4, 7]. Thus, shared spaces, particularly indoor spaces with higher respiratory activity including eating and talking, pose increased risk for individuals to inhale SARS-CoV-2.

With this increased risk accompanying previously very typical activities have come efforts to model aerosol and droplet concentrations in various settings including indoor dining, hospitals, planes, and classrooms [8–11]. While particles come in a continuum of sizes, there is a common, though at times problematic, attempt to classify them as either droplets, the larger particles that settle out of air relatively quickly (e.g., typically within 2m from being exhaled), or aerosols, the smaller particles that can stay airborne for hours. There has been vigorous debate about how much of a role aerosols play in the transmission of SARS-CoV-2, but the evidence has become clear that aerosols are the dominant transmission mode in most indoor settings [3, 12–14]. We focus on only aerosol-based transmission in this paper.

Dose response modeling is one method for translating aerosol virion concentration to individual-level probability of infection [15, 16]. Evans [17] presents a relatively simple mathematical model for COVID-19 aerosol exposure assuming a well-mixed room. Watanabe [16] provides an exponential dose response function formulated for SARS Coronavirus. Dabisch et al. [18] recently estimated a dose-response function for SARS-CoV-2 and found an exponential dose response function and parameters in rough agreement with those from [16]. We build from this previous work, particularly that of Evans [17] and Watanabe [16] to model how individuals interact with aerosols in an enclosed space under various conditions and for various activities and the infections that result. These efforts present opportunities to evaluate the effects of interventions such as air purifiers (e.g., HEPA filtration and ultraviolet light),

improved HVAC systems, and masking on virion concentration and health outcomes in a spaces ranging from a single room to a college [8, 9, 12, 19].

As universities, schools, and communities grapple with the risks associated with both in-person and virtual learning, researchers have taken advantage of simulation methods to formulate of the spread and impact of COVID-19 that accompanies in-person interactions, from entire states [20] to universities [21–23], to K-12 schools [11, 24, 25], and individual rooms [26]. Such endeavors take advantage of compartmental modeling, agent-based modeling, and social network analysis methods to evaluate interventions that include testing, mask-wearing, isolation, and distancing measures. In general, agent-based models (ABMs) account for complex dynamics of individual agent-level decisions for a particular community where data is available or results are aggregated enough to reflect the wide range of possible interactions. But, they are not easily adaptable across community populations or facilities. ABMs are useful for simulating interactions based on prototypical activity spaces across a state [20] or campus [22, 23, 26], including classrooms, dormitories, libraries, office buildings, and restaurants. In these cases, ABMs may be helpful for evaluating high-level, broad policy interventions such as distancing, masking, and surveillance testing. Conversely, ABMs can also be useful for evaluating risk drivers in single spaces (e.g., classrooms) over short time horizons where interactions and fluid dynamics can be represented, such as in [26]. However, ABMs are not suited for evaluating individual room or facility-level interventions over longer time horizons (e.g., a semester) because of the computational challenges and the difficulties capturing behavior and room-level details over these longer horizons. Compartmental SEIR models, on the other hand, rely on simulating transmission through close contacts between infectious and susceptible individuals, which can take advantage of social network modeling and differentiating between shared classrooms versus households to show population level impacts of COVID-19 from [11, 24, 25]. But, compartmental models must assume room parameters for defining close contacts rather than capturing varying risk of different spaces or non-linear risk of multiple infectious individuals in a shared room [27].

Our simulation model is unique from these approaches in that we utilize a risk analysis framework for assessing the uncertainty accompanying both COVID-19's parameters and potential policy responses. In addition, while still allowing for tracking susceptible, exposed, infectious, and removed individuals over the course of a college semester, our method differs from compartmental models in our ability to model individual class meeting room aerosol concentrations. Thus, our method enables evaluation of both individual room- and human-level interventions rather than relying on applying a single basic reproductive number, $R_0$, to larger populations or an average number of daily contacts [21, 23–25]. Furthermore, our model stands out from other university agent-based models as we incorporate actual individual classroom facility data, including volume and airflow, rather than assuming universal classroom size or sampling from prototypical classrooms. Consistent across all of these methodologies is the demonstrated effectiveness of defense in depth, where the compounding effects of multiple interventions are necessary for maintaining manageable levels of transmission risk [20, 25, 26] and the challenges associated with successfully translating simulation model results into adaptive decision-making [21–23].

# Methods

## Ethics statement

This project and date therein received approval of exemption by the University of Michigan's Health Sciences and Behavioral Sciences Institutional Review Board under the ID

HUM00206822. This data was approved for the purpose of secondary research and so was deemed exempt from informed consent of the subjects involved.

## General approach

To understand the risks of COVID-19 in the context of in-person classes on a college campus, we developed a simulation model to assess the probability of aerosol transmission of COVID-19 and the associated adverse health outcomes for students and faculty. We structured this simulation into three layers: class period, day, and semester. After initializing infection status at the beginning of the semester, we assess students' daily exposure to SARS-CoV-2 aerosol particles in the classroom. Given a single day's cumulative viral exposure, we apply a dose-response function [16] to assess an individual's probability of developing COVID-19 each day. We inject additional exogenous infections at the end of each day based on prevalence of COVID-19 in the community. We simulate every class meeting on every day over a period of 13 weeks to represent a semester to determine the total number of in-class infections for non-immune students. We used actual student and faculty class schedules from our example college as input to demonstrate the model. For each in-class infection during the semester, we simulate the risk of adverse health outcomes (hospitalization, death) based on age-specific rates from the health department of the county that is home to our example college.

This simulation starts with an input schedule array with all students and faculty assigned to classes and all classes assigned to rooms based on Fall 2019 (the last pre-pandemic fall semester) data provided by the college. Each individual is randomly assigned immunity status (defined as prior infection or vaccination) based on user-set immunity rate parameters for students and faculty. This reflects the combined influence of prior infection and vaccination. Individuals are then randomly assigned a weekday testing day to implement a weekly testing policy, reflecting the college's testing policy at the time this model was created. Initial infections at the beginning of the semester are randomly assigned to non-immune individuals based on a user-set parameter. The length of infectiousness of each infection is assigned based on a process described later in this paper that includes lag time, testing policy, and asymptomatic rate. Finally, we initialize arrays to track the days when individuals become infected and when they are actively infectious.

For each day and for each class that meets that day, the number of infected persons and the air changes per hour (ACH) of the classroom are used to simulate a non-immune individual's in-class exposure to SARS-CoV-2 assuming a well-mixed, equilibrium concentration of SARS-CoV-2 containing aerosol particles in the room using [17] as described in S1 Appendix. Note, at the time of publication this formulation from [17] was not yet peer reviewed, but the equation utilized for calculating cumulative exposure is verifiable by unit calculations and could be replaced in future applications of the model if appropriate. At the end of each day, we sum each non-immune individual's total daily exposure. We apply an exponential dose-response function based on [16] to determine their probability of developing COVID-19 on that day and randomly assign infectiousness as a Bernoulli trial. We incorporate exogenous infections by assigning infectiousness to the remaining non-immune and uninfected individuals based on a user-defined external infections rate. This process is repeated for each day of the semester.

At the end of the semester, we identify all students infected in class over the semester (that is, removing initial exogenous infections from the total). For each in-class infection, we simulate hospitalizations based on age-specific hospitalization rates from the local county public health records. For each resulting hospitalization, we simulate death again based on age-

specific rates consistent with county data. We replicate each semester 1000 times, beyond what we identified as necessary to achieve stochastic convergence of the simulation model.

## Data

Schedule data was provided by the college based on the Fall 2019 student class schedule and the Fall 2019 course schedule with room assignments. The Fall 2019 data initially include 15,221 students and 989 instructors across 2,280 classes. The course schedule contained 1,067 classes and their room and time assignments. Many of the classes include independent study and virtual classes that are not assigned to physical spaces, so we included only those courses with regular in-person meetings. Further, many of the courses contain multiple instructors, as the data set includes teaching assistants and faculty members as course instructors. The college also provided age ranges for faculty members. We cross referenced these faculty ages with the class schedule to separate faculty from student instructors. Ultimately, the schedule input for the demonstration of the model included 11,968 students and 342 faculty in 1,025 courses.

The course schedule data also contains room specifications critical for exposure calculations including the classroom volume and the length of class period. Some rooms contained airflow parameters. However we noticed inconsistencies in stated versus measured airflow for a sample of classrooms, and so assigned a reasonable estimate of 3.5 air changes per hour (ACH) in each room. We subsequently measured ACH values in a subset of the rooms. While some were higher, others were lower. This parameter could easily be updated if specific and accurate room ACH values are known.

As SARS-CoV-2 impacts vary by age, the health risks born by students versus faculty also vary. These differences should be reflected in modeling outcomes. We incorporate age-specific hospitalization and death rates from the county health department COVID-19 data collection for the county the college is in. Table 1 below shows values for rates of hospitalization given infection and of death given hospitalization by age group as calculated from the county data.

The data from the county for the 18–24 age group is, however, insufficient. There were no deaths in this age group, yet we know the risk is not zero. At the same time, there were substantially fewer college-age students in the county during much of the period covered by this data because the college was largely virtual. This poses a challenge. Rather than using the county data, we set the death rate for this age group to 1.90% to match the death rate in the next highest category. This parameter could be adjusted for future implementations of the model.

## Parameters and assumptions

The model makes some foundational assumptions about COVID-19 transmission. Foremost, the model considers immunity from vaccination and prior natural infection to provide perfect immunity for the duration of the semester. We know this is not true, particularly given the

**Table 1. COVID-19 hospitalization (given infection) and death (given hospitalization) rates by age in the county in which our example college is located.** This data is taken from a publicly available county web page.

| Age | Hospitalization | Death |
|---|---|---|
| 18–24 | 0.60% | 0.00% |
| 25–39 | 2.73% | 1.90% |
| 40–49 | 5.46% | 1.90% |
| 50–59 | 8.73% | 7.11% |
| 60–69 | 15.01% | 13.27% |
| 70+ | 34.30% | 36.97% |

recent surge in the Delta variant with associated break-through infections. This likely underestimates the risk, particularly of infection. We are extending this aspect of the model to account for imperfect vaccination in ongoing work. While students can become infected (and thereby gain immunity) mid-semester, the model assumes that no additional vaccinations occur in the student body after the start of the semester. The model solely represents aerosol transmission and does not consider transmission from fomites (inanimate objects such as surfaces or floors) or droplets. Each room is assumed to be well-mixed, such that each occupant receives an equal dose of viral particles. External infections can be configured to be constant over time (implying consistent local area spread) or time-varying (implying spikes); our demonstration runs assumed a constant external infection parameter.

To determine the length of an individual's infectiousness, we assume a somewhat-simplified dynamic for infectivity in which individuals become infectious two days after being infected with SARS-CoV-2. Non-immune students are tested for COVID-19 weekly on a fixed day-of-week based on an assigned testing day at the beginning of the simulation. Testing is assumed to be perfectly accurate and students are assumed to comply with quarantine measures (i.e., not attend class) after testing positive, though we include a one-day lag in receiving test results. We acknowledge that neither of these simplifying assumptions are strictly true for all students; testing is not perfect and experience has shown that students do not perfectly follow quarantine rules. Infected students are classified as either symptomatic (who leave the classroom one day after becoming infectious), or asymptomatic (who are removed one day after testing positive, accounting for the testing lag).

Given the recency of the SARS-CoV-2 pandemic, there is still great uncertainty in the nature of this virus and the details of its spread. In particular, while experts hold some consensus in modeling dose response for COVID-19 as exponential, the parameter value of this exponential function is still unknown. However, including and communicating this uncertainty is critical to evaluating risk. To incorporate this uncertainty, we bound the probability of infection based on an exponential dose response function $P(Infection) = 1 - exp^{(-dose/k)}$ with $k$ values selected to represent the lowest and highest expected transmissibility ($k = 500$ and $k = 75$, respectively) based on values from [16].

## Model functions

The simulation model and the functions within it are described below in Fig 1.

In a single class, we identify the number of infected individuals and the number of infectable individuals from the active infections tracking table. If these values are both greater than zero, we use a well-mixed room function to determine class-goer's aerosol exposure. Classroom parameters include room area, height, airflow (ACH), and the length of the class meeting period. We then calculate the in-class exposure $N_A$ based on equation 8 from Evans [17] below where breathing at rate $r_b$ in a room with aerosol virion concentration $\rho_A$ cumulative exposure $N_A$ in a room is proportional to the time spent in the room $t$:

$$N_A = \rho_A \, r_b \, t \tag{1}$$

The aerosol source rate is calculated as a function of the number of infected students and faculty in the room, with different source rates representing the variability in aerosol escape rate expected from an observing student versus a lecturing faculty member. This algorithm is shown in detail in Algorithm 1 in S2 Appendix with model formula details provided in S1 Appendix.

At the end of each simulated day, we sum each non-immune individual's aerosol exposure over all of their class meetings in a day. We calculate a probability of infection from that

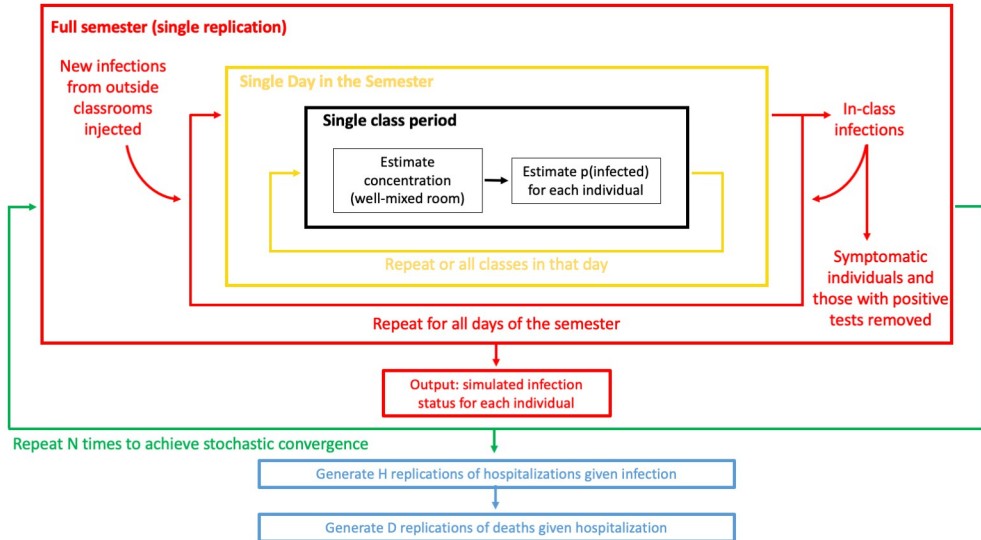

**Fig 1. Overview of simulation model.**

exposure based on an exponential dose response function parameterized with low and high transmissibility values as described above. This function is detailed in Algorithm 2 of S2 Appendix. Any changes in infection states are updated in the tracking infections table.

For each infection, we further simulate the length of infectiousness. We assign symptomaticity as a Bernoulli trial based on the user-input asymptomatic rate. For symptomatic individuals, we assign infectiousness for a single day after the lag period based on the assumption that students will stop going to class when they have symptoms, as college policy and required daily symptom checking dictate. For asymptomatic individuals, we assign infectiousness for all days between the lag period and a day after the individual's next test day, accounting for delay in receiving test results. The results of this function update the active infections table.

**Table 2. User-settable parameters for modeling transmission of and outcomes from COVID-19 infection.**

| Parameter | Value | Source for parameter values |
|---|---|---|
| initial semester infection prevalence among students and faculty | 1% | Expert judgement on local community prevalence |
| infections introduced from outside the classroom daily among students and faculty | 0.2% | weekly college testing data |
| incubation period | 2 days | [22] |
| asymptomatic rate | 40% | [11] |
| viral particle settling time ($\tau_{settle}$) | 20 min | [17] |
| viral deactivation time ($\tau_{deact}$) | 90 min | [17] |
| viral load in saliva ($\rho_0$) | 1000 /nL | [17] |
| breathing rate ($r_b$) | 10 L/min | [17, 28] |
| student aerosol source rate ($r_{stud}$) | 1 nL/min | [29] |
| faculty aerosol source rate ($r_{fac}$) | 5 nL/min | [29] |

At the end of the semester, we simulate two possible adverse health outcomes of infection: hospitalization given infection and death given hospitalization. For each individual infected in class during the semester, we simulate 100 replications of hospitalization versus not hospitalized with a probability of hospitalization given infection corresponding to the age of the infected individual as shown in Table 1 above. For each simulated hospitalization, we simulate death or not given hospitalization with probability based on age groups from Table 1. As these probabilities are much smaller, we simulate 1000 iterations per hospitalization. These simulations provide probability distributions for the cumulative number of hospitalizations and deaths for a given scenario as well as the risk of hospitalization and death for each individual. This algorithm is detailed in Algorithm 3 in S2 Appendix.

### Scenario development

The model considers a set of parameters related to the spread dynamics of SARS-CoV-2. The parameters we set but keep constant in our demonstration are described in Table 2. These parameters are meant to correspond to the situation at our example college at the time this model was created (April and May 2021). These input parameters are designed to be easily changed to adapt to model users and their community conditions.

Evaluating the relative risks associated with in-class meetings for the semester includes exogenous uncertainties around immunity as well as possible facility and policy interventions. Immunity rates in the fall were still uncertain at the the time this model was created. To separate the impacts of masking and UVC-fan interventions from the range of possible immunity rates, we simulate over varying college immunity rates from 60% to 95%.

While we could include innumerable interventions, for the purposes of demonstration in this paper we limit our consideration to masking mandates and the installation of UVC ceiling fans that increase the rate of aerosol deactivation and increasing mixing in the room to counter stagnant areas. To determine the effect of UVC fans, we calculate an equivalent viral decay rate based on parameters provided by the UVC fan manufacturers (S1 Appendix) consistent with literature on inactivation models for germicidal UVC [30, 31]. This is not meant to be the definitive assessment of the decay parameters for any UVC intervention, particularly since precise inactivation data on aerosols of SARS-CoV-2 are not yet published in peer reviewed literature. Rather, these parameters give us a reasonable, even conservative, starting point for our demonstration of structural environmental interventions and are based on experiments with live SARS-CoV-2 virus on surfaces [32, 33], other coronaviruses as aerosols [34, 35], and recommendations from the American Society of Heating, Refrigeration, and Air-Conditioning Engineers [36, 37]. This decay rate is added to the other viral decay and settling rates that ultimately factor into the aerosol concentration decay variable. We implement the effectiveness of masking on aerosols by applying a masking coefficient to the aerosol source rates $r_{stud}$ and $r_{fac}$. We also apply a masking coefficient to the inhalation rate ($r_b$), the same for students and faculty. Ultimately, 20 scenarios are evaluated across five different immunity rates (60%, 70%, 80%, 90%, 95%), with and without masking, and with and without UVC fans.

## Results

### Scenario comparison

For each individual in each of the scenario combinations, we calculate the probability of in-class infection of a non-immune individual during the full semester by dividing the total number of replications in which they were infected in that scenario by the total number of replications where that individual was not assigned immunity in that scenario. We then generate distributions based on the non-immune individuals' probability of infection over the evaluated

scenarios. This then represents the probability distribution of individual probabilities of infection taken over the individuals in the college. This captures differences in likelihood of infection due to course schedules (e.g., a student with one small course vs. one with 4 large courses) and room properties. We also compute the college-cumulative number of infections, hospitalizations, and deaths. We do this by summing each of these quantities over all students and all faculty for each replication of each scenario. Together this process yields the information needed to estimate probability density functions over the individual probabilities of each outcome (infection, hospitalization, and death) and the probability density functions over the total number of each outcome (infection, hospitalization, and death) in the college. These support different decisions. The individual-level probability estimates support decisions by both individuals and college leadership about whether or not risk levels are acceptable at the individual level under different mitigation options. The college-total estimates are aimed primarily at supporting college-level decision-making about which mitigation options to implement and the logistical and administrative impacts of these decisions.

While the true probability of infection in the fall 2021 semester cannot be precisely predicted given the uncertainties inherent to COVID-19 and to simulation in general, the low-transmissibility and high-transmissibility distributions provide bounds on infection probabilities under the given conditions and assumptions. Stating this differently, neither of these curves likely is the true risk curve for the college. The true risk curve is likely somewhere between these bounds if our assumptions are reasonable. For a more transmissible variant it is likely towards the upper end. For a less transmissible variant it is likely near the lower end. From these we can compare differences in risk when decision-makers introduce interventions.

Figs 2 and 3 show the relative risk of in-class aerosol infection comparing scenarios with and without masking for students (Fig 2) and faculty (Fig 3) with a 95% immunity rate. The graphs show the model results in the form of inverse cumulative density functions. These show the probability of exceeding a given number of infections. Full results for student and faculty infections including other immunity rates and scenarios are shown in S3 Appendix.

Consider first the results for student infections for the high transmissibility scenarios in Fig 2. The dashed lines show that if there is 95% immunity and no masking, there is a probability of about 0.943 of exceeding 50 infected students. If masking is added, the probability of exceeding 50 infections drops to about 0.008. Similar figures for student infection inverse cumulative density functions across all the tested interventions and immunity rates scenarios are presented in S3 Appendix. We list the probability of college infections exceeding example infection levels of 50 and 100 students in Table 3.

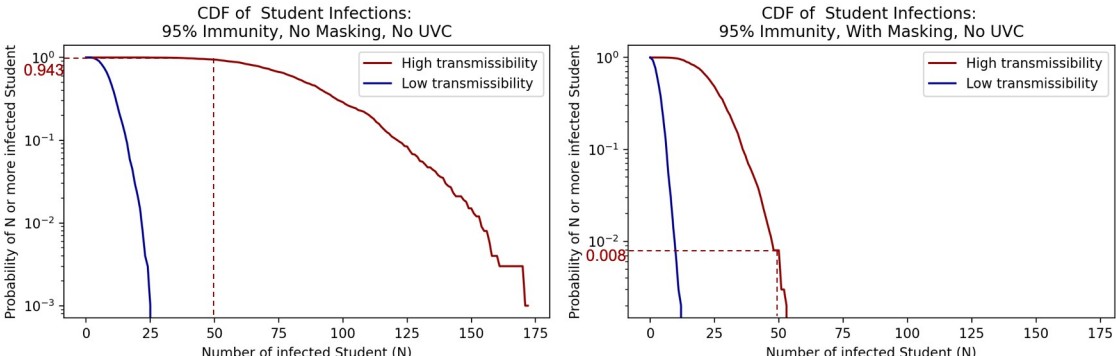

**Fig 2. Inverse cumulative density functions for the total number of students infected under the different scenarios.** Each graph shows the curve of the probability of exceeding a given number of infections.

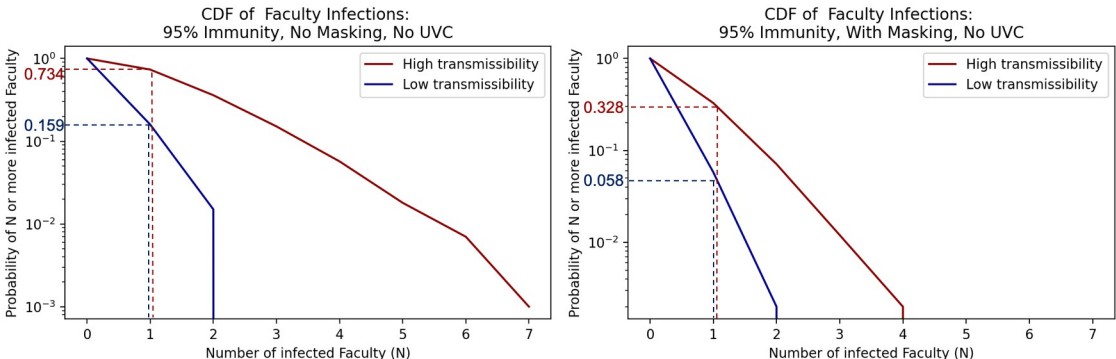

**Fig 3. Inverse cumulative density functions for the total number of faculty infected under the different scenarios.** Each graph shows the curve of the probability of exceeding a given number of infections.

Table 3 shows that, given the model assumptions and without interventions, if immunity in the community is instead 90%, there is near certainty (>0.999) of exceeding 100 infections and a probability of 0.586 of exceeding 500 student infections in the high transmissibility case. If masking is adopted with 90% immunity, the probability of exceeding 100 and 500 infections drop to 0.914 and <0.001 respectively. At the lowest immunity value used in this study, 60%, the probabilities of exceeding 100 and 500 student infections are approximately 1 both without and with masking. Masking alone cannot counteract low immunity rates in this setting. It is only with the addition of both masking and UVC with 60% immunity that the probabilities of exceeding 500 student infections drops to 0.554 (the probability of exceeding 100 infections is still approximately 1).

Consider next the results for faculty infections for the high transmissibility scenarios. Fig 3 shows that for the 95% immunity case the probabilities of exceeding 1 faculty infection are between the low transmissibility and high transmissibility bounds of 0.159 and 0.734 without masking. With masking, these probabilities shift down to between 0.058 for the low transmissibility case and 0.328 for the high transmissibility case. Corresponding figures for faculty infection inverse cumulative density functions across all scenarios are presented in S3 Appendix. We list the probability of cumulative infections exceeding various infection levels for faculty in Table 4.

Using Table 4 if we consider the 90% immunity scenario, the probabilities of exceeding 5 and 20 faculty infections without masking are 0.9 and <0.001 under the high transmissibility case. If masking is added, the probability of exceeding 5 infections drops to 0.109 for the high transmissibility case. If only UVC is used (without masking) this probability is instead 0.008. If both masking and UVC are employed, this probability drops to <0.001. If we instead look at the 60% immunity scenarios, the probabilities of exceeding 5 and 20 faculty infections are approximately 1 both without and with masking in the high transmissibility case. Only if both masking and UVC are implemented do these high transmissibility case probabilities drop to 0.926 and 0.005.

From these tables and their corresponding figures in S3 Appendix, we see that the use of masks significantly reduces the probability of infection for non-immune students and faculty, consistent with international policy recommendations, guidelines, and mandates in place over the pandemic period for indoor gatherings. In addition, the introduction of UVC fans in every classroom reduces the risk of infection more than universal masking alone, and UVC fans are not as directly dependent on behavioral compliance as masking. Finally and as expected, masking and UVC fans used in combination show the greatest reduction in risk compared to a

**Table 3. Probability of exceeding various student infection levels.**

| Immunity rate | Masking | UVC Fans | P(Infections ≥ 50) | | P(Infections ≥ 100) | | P(Infections ≥ 250) | | P(Infections ≥ 500) | |
|---|---|---|---|---|---|---|---|---|---|---|
| | | | Low Transmissibility | High Transmissibility | Low Transmissibility | High Transmissibility | Low Transmissibility | High Transmissibility | Low Transmissibility | High Transmissibility |
| 60% | No Masking | No UVC | >0.999 | >0.999 | >0.999 | >0.999 | >0.999 | >0.999 | >0.999 | >0.999 |
| | With Masking | No UVC | >0.999 | >0.999 | >0.999 | >0.999 | 0.956 | >0.999 | <0.001 | >0.999 |
| | No Masking | With UVC | >0.999 | >0.999 | 0.997 | >0.999 | <0.001 | >0.999 | <0.001 | >0.999 |
| | With Masking | With UVC | 0.313 | >0.999 | <0.001 | >0.999 | <0.001 | >0.999 | <0.001 | 0.554 |
| 70% | No Masking | No UVC | >0.999 | >0.999 | >0.999 | >0.999 | >0.999 | >0.999 | 0.998 | >0.999 |
| | With Masking | No UVC | >0.999 | >0.999 | 0.994 | >0.999 | 0.001 | >0.999 | <0.001 | >0.999 |
| | No Masking | With UVC | 0.994 | >0.999 | 0.066 | >0.999 | <0.001 | >0.999 | <0.001 | >0.999 |
| | With Masking | With UVC | 0.001 | >0.999 | <0.001 | >0.999 | <0.001 | 0.368 | <0.001 | <0.001 |
| 80% | No Masking | No UVC | >0.999 | >0.999 | >0.999 | >0.999 | 0.412 | >0.999 | <0.001 | >0.999 |
| | With Masking | No UVC | 0.853 | >0.999 | 0.002 | >0.999 | <0.001 | >0.999 | <0.001 | >0.999 |
| | No Masking | With UVC | 0.014 | >0.999 | <0.001 | >0.999 | <0.001 | 0.991 | <0.001 | 0.03 |
| | With Masking | With UVC | <0.001 | 0.993 | <0.001 | 0.303 | <0.001 | <0.001 | <0.001 | <0.001 |
| 90% | No Masking | No UVC | 0.281 | >0.999 | >0.999 | >0.999 | <0.001 | >0.999 | <0.001 | 0.586 |
| | With Masking | No UVC | <0.001 | 0.999 | <0.001 | 0.914 | <0.001 | 0.003 | <0.001 | <0.001 |
| | No Masking | With UVC | <0.001 | 0.814 | <0.001 | 0.034 | <0.001 | <0.001 | <0.001 | <0.001 |
| | With Masking | With UVC | <0.001 | <0.001 | <0.001 | <0.001 | <0.001 | <0.001 | <0.001 | <0.001 |
| 95% | No Masking | No UVC | <0.001 | 0.943 | <0.001 | 0.288 | <0.001 | <0.001 | <0.001 | <0.001 |
| | With Masking | No UVC | <0.001 | 0.008 | <0.001 | <0.001 | <0.001 | <0.001 | <0.001 | <0.001 |
| | No Masking | With UVC | <0.001 | <0.001 | <0.001 | <0.001 | <0.001 | <0.001 | <0.001 | <0.001 |
| | With Masking | With UVC | <0.001 | <0.001 | <0.001 | <0.001 | <0.001 | <0.001 | <0.001 | <0.001 |

**Table 4. Probability of exceeding various faculty infection levels.**

| Immunity rate | Masking | UVC Fans | P(Infections ≥ 1) | | P(Infections ≥ 5) | | P(Infections ≥ 10) | | P(Infections ≥ 20) | |
|---|---|---|---|---|---|---|---|---|---|---|
| | | | Low Transmissibility | High Transmissibility | Low Transmissibility | High Transmissibility | Low Transmissibility | High Transmissibility | Low Transmissibility | High Transmissibility |
| 60% | No Masking | No UVC | >0.999 | >0.999 | >0.999 | >0.999 | 0.999 | >0.999 | 0.887 | >0.999 |
| | With Masking | No UVC | 0.993 | >0.999 | 0.67 | >0.999 | 0.091 | >0.999 | <0.001 | >0.999 |
| | No Masking | With UVC | 0.936 | >0.999 | 0.156 | >0.999 | 0.002 | >0.999 | <0.001 | >0.999 |
| | With Masking | With UVC | 0.589 | >0.999 | 0.004 | 0.926 | <0.001 | 0.38 | <0.001 | 0.005 |
| 70% | No Masking | No UVC | >0.999 | >0.999 | 0.992 | >0.999 | 0.772 | >0.999 | 0.033 | >0.999 |
| | With Masking | No UVC | 0.91 | >0.999 | 0.182 | >0.999 | 0.003 | >0.999 | <0.001 | 0.999 |
| | No Masking | With UVC | 0.743 | >0.999 | 0.026 | >0.999 | <0.001 | 0.992 | <0.001 | 0.455 |
| | With Masking | With UVC | 0.382 | 0.987 | <0.001 | 0.399 | <0.001 | 0.018 | <0.001 | <0.001 |
| 80% | No Masking | No UVC | 0.977 | >0.999 | 0.379 | >0.999 | 0.011 | >0.999 | <0.001 | 0.997 |
| | With Masking | No UVC | 0.674 | >0.999 | 0.006 | 0.997 | <0.001 | 0.916 | <0.001 | 0.092 |
| | No Masking | With UVC | 0.435 | 0.998 | 0.002 | 0.728 | <0.001 | 0.132 | <0.001 | <0.001 |
| | With Masking | With UVC | 0.196 | 0.774 | <0.001 | 0.039 | <0.001 | <0.001 | <0.001 | <0.001 |
| 90% | No Masking | No UVC | 0.497 | >0.999 | 0.003 | 0.904 | <0.001 | 0.339 | <0.001 | <0.001 |
| | With Masking | No UVC | 0.224 | 0.881 | <0.001 | 0.109 | <0.001 | <0.001 | <0.001 | <0.001 |
| | No Masking | With UVC | 0.122 | 0.652 | <0.001 | 0.008 | <0.001 | <0.001 | <0.001 | <0.001 |
| | With Masking | With UVC | 0.056 | 0.266 | <0.001 | <0.001 | <0.001 | <0.001 | <0.001 | <0.001 |
| 95% | No Masking | No UVC | 0.159 | 0.734 | <0.001 | 0.018 | <0.001 | <0.001 | <0.001 | <0.001 |
| | With Masking | No UVC | 0.058 | 0.328 | <0.001 | <0.001 | <0.001 | <0.001 | <0.001 | <0.001 |
| | No Masking | With UVC | 0.043 | 0.204 | <0.001 | <0.001 | <0.001 | <0.001 | <0.001 | <0.001 |
| | With Masking | With UVC | 0.016 | 0.074 | <0.001 | <0.001 | <0.001 | <0.001 | <0.001 | <0.001 |

no-intervention scenario. With the decreases in infection rates we also see a narrowing of uncertainty bounds, indicating decreases in uncertainty offered by these interventions. These figures and tables also highlight the importance of high rates of immunity (vaccination plus immunity due to prior infection) in the community. Masking or UVC alone cannot overcome the vulnerabilities caused by lower immunity rates. Only when masking and UVC are used together do we begin to see reduced transmission in the low immunity rate scenarios.

Fig 4 shows the cumulative density function over the probability of infection at the level of individual students and faculty for those who are not vaccinated when 95% of the population is immune for students and faculty. These plots allow a decision maker to determine what fraction of the population is above a given level of individual risk (e.g., a threshold or acceptable risk). With no masks and no UVC, the median (0.50 on the y-axis) probability of infection is between <0.001 and 0.128 for students and between <0.001 and 0.052 for faculty. In other words, if a student or faculty member were drawn at random from the college, half would have a probability of infection at or below the median level and half would have a probability of infection above the median level, reflecting differences in course schedules and classrooms. With masking added the median probability of infection for students is between <0.001 and 0.033 and for faculty the median probability of infection with masking added is between <0.001 and 0.016. Adding UVC alone reduces these probability more than masking alone

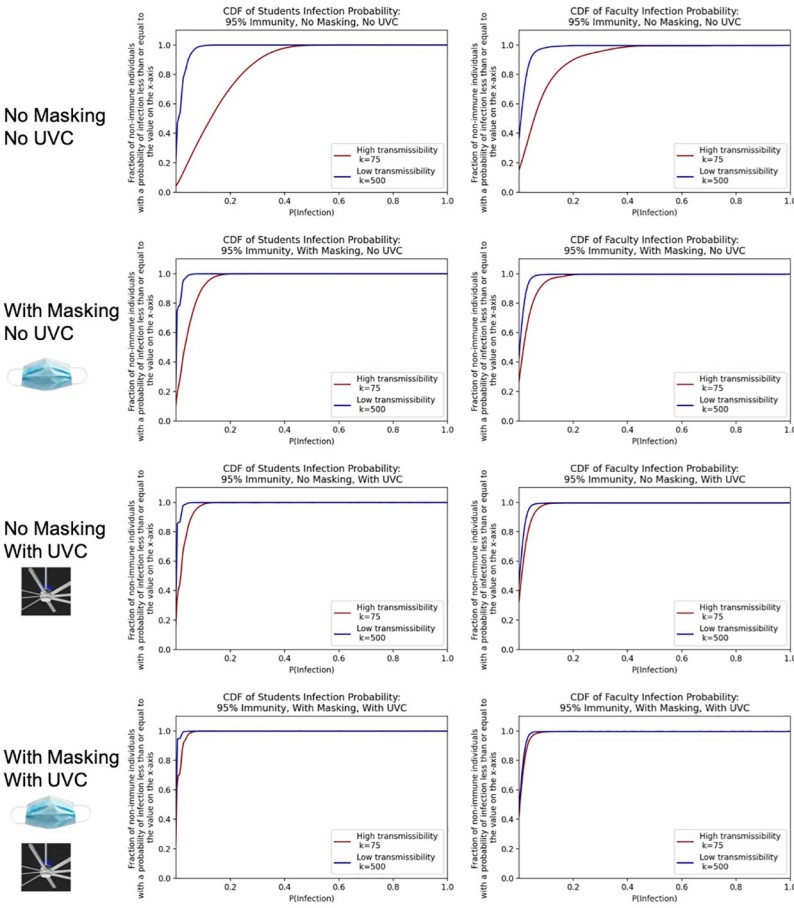

**Fig 4. Cumulative density functions over the probabilities of individual student and faculty infection given 95% immunity in the college.**

(between <0.001 and 0.019 for students, <0.001 to <0.001 for faculty). Together UVC and masking results in further reductions relative to each separately.

If immunity rates were instead 90% (results shown in S4 Appendix) the median probability of infections for students without masking is between 0.029 and 0.426, and the median probability of infection for faculty is between 0.011 and 0.195. If masking is added the median range for students drops to 0.01 to 0.10 and the median bounds for faculty drops to <0.001 to 0.042. If UVC fans were added rather than masking with 90% immunity, the student median infection probability is between <0.001 to 0.044 and the faculty median infection probability is between <0.001 and 0.019. These results show clearly that masking and UVC fan interventions diminish both the probability of infection as well as the range of uncertainty due to the different possibility transmissibility $k$ values. Similar plots showing student and faculty probabilities of infection for other immunity rates are also in S4 Appendix.

Fig 5, similar to that featured in [21], summarizes the average number of modeled classroom-acquired infections across scenarios and transmissibility $k$ values. Adopting a policy that employs both UVC fans and universal masking even in the highest transmissibility scenario, shown by the solid red line with circle markers, reduces the average number of infections more than doing nothing in a low transmissibility scenario, shown by the blue dashed line with diamond markers.

We evaluated health outcomes including hospitalization and death based on the county health department rates described in the Data section. We show associated probabilities for at

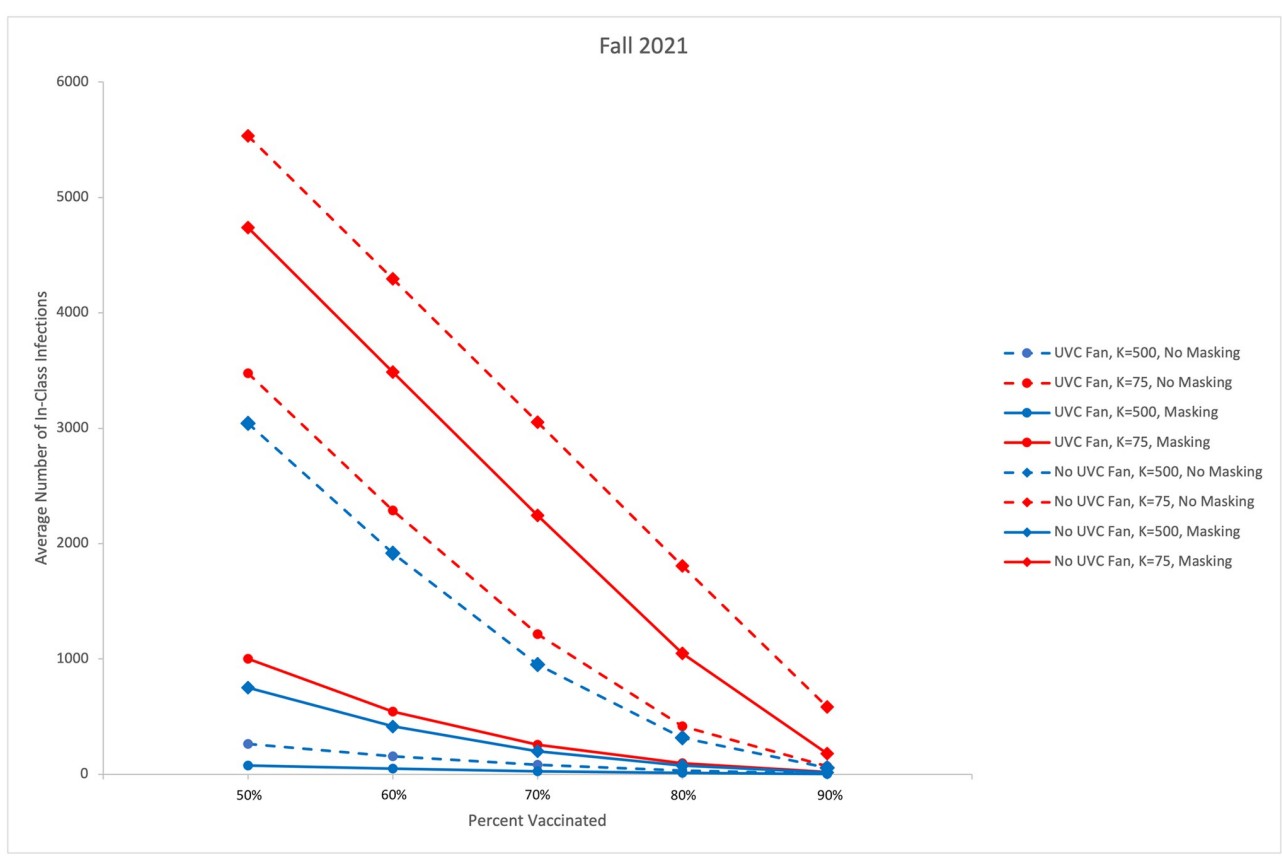

**Fig 5. Average infections by immunity rate and scenario.**

least a number (N) of student hospitalizations in Fig 6 or deaths in Fig 7 under 95% immunity rates for the no UVC scenarios with and without masking.

Consider the scenario with 95% immunity and no masks in Figs 6 and 7. Given the assumptions and uncertainties underlying the model, the probability of at least 1 student being hospitalized due to transmission in the classroom is estimated to be between 0.055 and 0.398 (low transmissibility and high transmissibility bounds respectively) in Fig 6. For this scenario the probability of having one or more deaths of students due to transmission in the classroom is between 0.001 and 0.01 shown in Fig 7. With universal masking and 95% immunity, the probability of 1 or more hospitalized students is estimated to be between 0.019 and 0.138, and the probability of 1 or more student deaths is estiamted to be between <0.001 for low transmissibility and 0.003 for high transmissibility.

We summarize results for different impact levels and varying masking and UVC use at all immunity rates in Table 5. If UVC fans are added to masking in the 95% immunity scenario, the probability of 1 or more hospitalized students is between 0.004 and 0.027, and the probability of 1 or more student deaths is between <0.001 and 0.005. From these results for the the 95% immunity scenario, we see masking reduces the probability of student hospitalization substantially and the probability of a student death by an order of magnitude. Adding UVC fans to universal masking reduces the probabilities of having one more student hospitalizations and one or more student deaths by an additional order of magnitude each.

Overall, Table 5 along with the corresponding plots in S3 Appendix show that increasing immunity levels (under the perfect immunity assumption) is effective for diminishing risk of hospitalizations and deaths in both low and high transmissibility specifications. However, even in the 95% immunity scenario, masking and UVC fans yield substantial further reductions in risk. Given uncertainty surrounding student immunity rates and the challenges faced by a college in enforcing a vaccination mandate, these plots also show how policy and facility interventions, individually or in combination, can contribute to mitigating risk.

Figs 8 and 9 similarly show the 95% community immunity rate probabilities of N or more hospitalizations or deaths for faculty members, who are in higher risk categories for negative health outcomes but perhaps hold fewer class hours than students. These same plots for all scenarios and immunity rates are compiled in S3 Appendix and summarized below for various impact levels in Table 6.

Consider again the 95% immunity scenario without masks or UVC. The probability of at least one faculty member being hospitalized is between 0.016 and 0.116, as shown in Fig 8, and the probability of at least one faculty death is between 0.002 and 0.018, as shown in Fig 9. If masks are added, the bounds on the probability of at least one faculty hospitalization drop to 0.005 to 0.040 as shown in Fig 8, and the bounds on the probability of at least one faculty death drop to <0.001 to 0.006 as shown in Fig 9. Both of these are an order of magnitude lower than without masks.

As with the hospitalization and death results for students, we see that for faculty high immunity rates effectively reduce risk. However, even at the high immunity rate of 95% immunity, the risk of hospitalizations and deaths is not zero. Masking and UVC further reduce the risk by an order of magnitude. If decision makers have a clear level of residual risk that they find acceptable, results such as these can help a decision maker determine if mitigation measures are needed and, if so, which ones to implement.

As hospitalizations and deaths vary by age, we can also show a distribution of any non-immune individual's probability of hospitalization and death outcomes by age. Below in Figs 10 and 11, we show these distributions for the 95% immunity rate scenario. Similar plots broken down by additional immunity rates can be found in S4 Appendix.

**Table 5. Probability of exceeding various student hospitalization and death levels.**

| Immunity rate | Masking | UVC Fans | P(Hospitalizations ≥ 1) | | P(Hospitalizations ≥ 10) | | P(Deaths ≥ 1) | | P(Deaths ≥ 5) | |
|---|---|---|---|---|---|---|---|---|---|---|
| | | | Low Transmissibility | High Transmissibility | Low Transmissibility | High Transmissibility | Low Transmissibility | High Transmissibility | Low Transmissibility | High Transmissibility |
| 60% | No Masking | No UVC | >0.999 | >0.999 | 0.450 | >0.999 | 0.162 | 0.375 | <0.001 | <0.001 |
| | With Masking | No UVC | 0.853 | >0.999 | <0.001 | 0.992 | 0.036 | 0.306 | <0.001 | <0.001 |
| | No Masking | With UVC | 0.587 | >0.999 | <0.001 | 0.839 | 0.017 | 0.220 | <0.001 | <0.001 |
| | With Masking | With UVC | 0.239 | 0.951 | <0.001 | 0.002 | 0.005 | 0.057 | <0.001 | <0.001 |
| 70% | No Masking | No UVC | 0.987 | >0.999 | 0.018 | 0.980 | 0.080 | 0.282 | <0.001 | <0.001 |
| | With Masking | No UVC | 0.610 | >0.999 | <0.001 | 0.753 | 0.018 | 0.203 | <0.001 | <0.001 |
| | No Masking | With UVC | 0.373 | 0.999 | <0.001 | 0.161 | 0.009 | 0.123 | <0.001 | <0.001 |
| | With Masking | With UVC | 0.140 | 0.759 | <0.001 | <0.001 | 0.003 | 0.027 | <0.001 | <0.001 |
| 80% | No Masking | No UVC | 0.760 | >0.999 | <0.001 | 0.561 | 0.027 | 0.175 | <0.001 | <0.001 |
| | With Masking | No UVC | 0.315 | 0.994 | <0.001 | 0.044 | 0.007 | 0.094 | <0.001 | <0.001 |
| | No Masking | With UVC | 0.176 | 0.899 | <0.001 | <0.001 | 0.004 | 0.044 | <0.001 | <0.001 |
| | With Masking | With UVC | 0.063 | 0.414 | <0.001 | <0.001 | 0.001 | 0.010 | <0.001 | <0.001 |
| 90% | No Masking | No UVC | 0.227 | 0.951 | <0.001 | 0.002 | 0.005 | 0.056 | <0.001 | <0.001 |
| | With Masking | No UVC | 0.081 | 0.564 | <0.001 | <0.001 | 0.002 | 0.016 | <0.001 | <0.001 |
| | No Masking | With UVC | 0.044 | 0.324 | <0.001 | <0.001 | <0.001 | 0.007 | <0.001 | <0.001 |
| | With Masking | With UVC | 0.015 | 0.110 | <0.001 | <0.001 | <0.001 | 0.002 | <0.001 | <0.001 |
| 95% | No Masking | No UVC | 0.055 | 0.398 | <0.001 | <0.001 | 0.001 | 0.010 | <0.001 | <0.001 |
| | With Masking | No UVC | 0.019 | 0.138 | <0.001 | <0.001 | <0.001 | 0.003 | <0.001 | <0.001 |
| | No Masking | With UVC | 0.011 | 0.076 | <0.001 | <0.001 | <0.000 | 0.001 | <0.001 | <0.001 |
| | With Masking | With UVC | 0.004 | 0.027 | <0.001 | <0.001 | <0.001 | <0.001 | <0.001 | <0.001 |

**Table 6. Probability of exceeding various faculty hospitalization and death levels.**

| Immunity rate | Masking | UVC Fans | P(Hospitalizations ≥ 1) | | P(Hospitalizations ≥ 10) | | P(Deaths ≥ 1) | | P(Deaths ≥ 5) | |
|---|---|---|---|---|---|---|---|---|---|---|
| | | | Low Transmissibility | High Transmissibility | Low Transmissibility | High Transmissibility | Low Transmissibility | High Transmissibility | Low Transmissibility | High Transmissibility |
| 60% | No Masking | No UVC | 0.915 | >0.999 | <0.001 | 0.455 | 0.325 | 0.792 | <0.001 | 0.020 |
| | With Masking | No UVC | 0.407 | 0.998 | <0.001 | 0.071 | 0.073 | 0.624 | <0.001 | 0.003 |
| | No Masking | With UVC | 0.219 | 0.973 | <0.001 | 0.004 | 0.036 | 0.444 | <0.001 | <0.001 |
| | With Masking | With UVC | 0.076 | 0.563 | <0.001 | <0.001 | 0.011 | 0.118 | <0.001 | <0.001 |
| 70% | No Masking | No UVC | 0.691 | 0.998 | <0.001 | 0.087 | 0.165 | 0.640 | <0.001 | 0.003 |
| | With Masking | No UVC | 0.223 | 0.968 | <0.001 | 0.002 | 0.036 | 0.426 | <0.001 | <0.001 |
| | No Masking | With UVC | 0.127 | 0.841 | <0.001 | <0.001 | 0.021 | 0.255 | <0.001 | <0.001 |
| | With Masking | With UVC | 0.039 | 0.323 | <0.001 | <0.001 | 0.005 | 0.056 | <0.001 | <0.001 |
| 80% | No Masking | No UVC | 0.316 | 0.959 | <0.001 | 0.002 | 0.056 | 0.405 | <0.001 | <0.001 |
| | With Masking | No UVC | 0.098 | 0.745 | <0.001 | <0.001 | 0.015 | 0.192 | <0.001 | <0.001 |
| | No Masking | With UVC | 0.053 | 0.450 | <0.001 | <0.001 | 0.007 | 0.088 | <0.001 | <0.001 |
| | With Masking | With UVC | 0.018 | 0.137 | <0.001 | <0.001 | 0.002 | 0.021 | <0.001 | <0.001 |
| 90% | No Masking | No UVC | 0.063 | 0.547 | <0.001 | <0.001 | 0.009 | 0.117 | <0.001 | <0.001 |
| | With Masking | No UVC | 0.024 | 0.198 | <0.001 | <0.001 | 0.004 | 0.033 | <0.001 | <0.001 |
| | No Masking | With UVC | 0.012 | 0.095 | <0.001 | <0.001 | 0.002 | 0.015 | <0.001 | <0.001 |
| | With Masking | With UVC | 0.005 | 0.029 | <0.001 | <0.001 | <0.001 | 0.004 | <0.001 | <0.001 |
| 95% | No Masking | No UVC | 0.016 | 0.116 | <0.001 | <0.001 | 0.002 | 0.018 | <0.001 | <0.001 |
| | With Masking | No UVC | 0.005 | 0.040 | <0.001 | <0.001 | <0.001 | 0.006 | <0.001 | <0.001 |
| | No Masking | With UVC | 0.003 | 0.022 | <0.001 | <0.001 | <0.001 | 0.004 | <0.001 | <0.001 |
| | With Masking | With UVC | 0.001 | 0.007 | <0.001 | <0.001 | <0.001 | 0.001 | <0.001 | <0.001 |

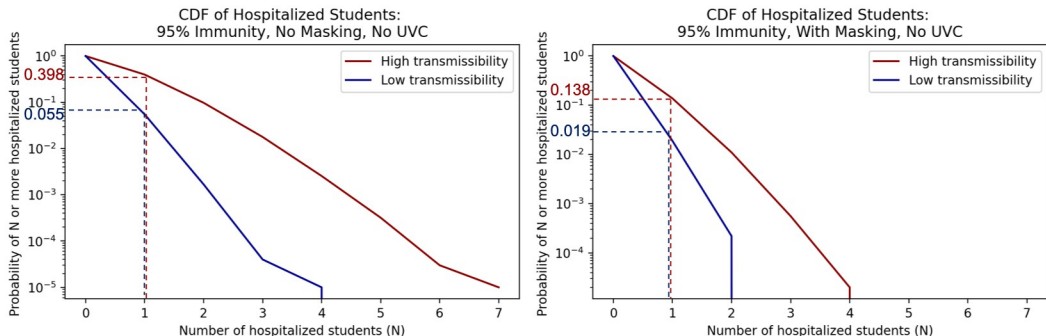

**Fig 6. Inverse cumulative density functions for the total number of students hospitalized under the different scenarios.** Each graph shows the curve of the probability of exceeding a given number of hospitalizations.

## Sensitivity analysis

We conducted sensitivity analyses on several parameters to better understand the impact of outside-class infections on in-class infections by testing different initial and exogenous (outside the classroom) infection rates. These sensitivity analyses all assume no masking or UVC, 70% vaccination rate, and a weekly testing policy.

We compared initial infection prevalence values of 0.5% and 2.0% with the base case value of 1.0% value used for the model runs as referenced in Table 2. Fig 12 shows that this parameter does not visibly shift the results of the model over the 13 week semester.

We next evaluated the daily exogenously introduced infections parameter set to 0.2% in the base case runs based on test positivity at the college in the most recent semester previous to the creation of this model. We tested the model with the daily exogenous infections set to 0.1% and 0.4%, which translates to about 0.5% and 2.0% weekly infections respectively. For these runs we kept the initial infection rate consistent with the original run of the model at 1.0%. From Fig 13 we see similar results in student infections across the range of 0.1% to 0.4% daily exogenous infections showing that the model is not sensitive to subtle differences in this parameter assumption that may come from errors in data around community infection prevalence. However, this range may not capture larger fluctuations in community infection prevalence from gatherings such as the start of the school year or large events like football games.

We also evaluated our assumption of 40% mask effectiveness to determine the range of masking's impact on the model results. The original 40% value included in the model was

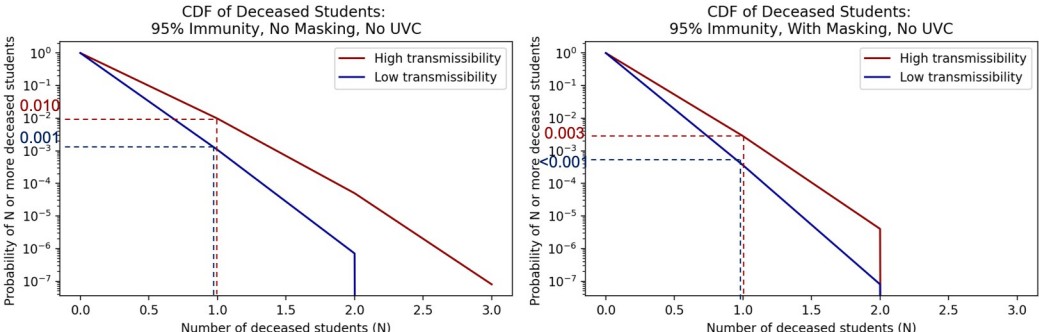

**Fig 7. Inverse cumulative density functions for the total number of students that die under the different scenarios.** Each graph shows the curve of the probability of exceeding a given number of deaths.

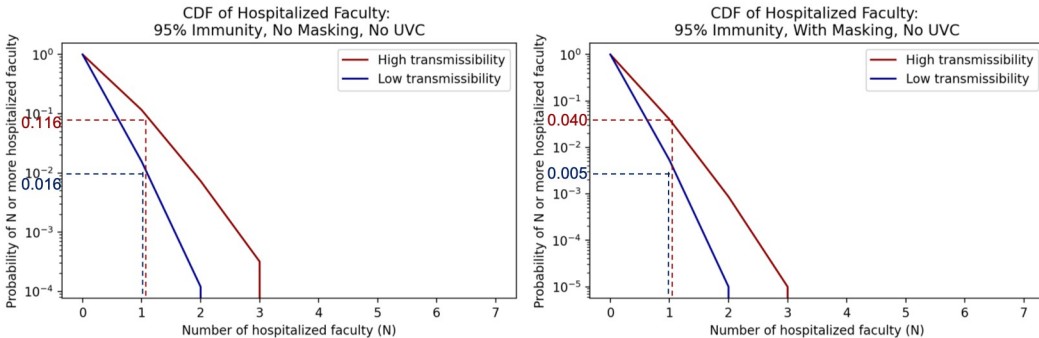

**Fig 8. Inverse cumulative density functions for the total number of faculty hospitalized under the different scenarios.**
Each graph shows the curve of the probability of exceeding a given number of hospitalizations.

based on expert judgement of the effectiveness of surgical masks worn as they commonly are in practice. In Fig 14 we compare these base case results to results with assumptions of 25%, 50%, 75%, and 95% mask effectiveness. This figure shows the large potential impact of masking and highlights the critical role that individual mask wearing can play on aerosol infection spread within a community. In particular, if an educational institution could ensure that those in the classroom, both students and faculty, wore high-quality masks **and wore them fitted correctly**, there is the potential for a larger reduction in risk than the 40% effectiveness of masks we assumed in our base case. Note, for these tests the initial infections parameter is set to 1.0% and the daily exogenous infections parameter is set to 0.2%, we do not include UVC, we assume a 70% vaccination rate, and a weekly testing policy.

As this model includes numerous other assumptions previously mentioned, future sensitivity runs could evaluate the impact of our other chosen default parameters such as asymptomatic rate, breathing rate, and room ACH. More in-depth sensitivity analysis could include testing assumptions around the uniform distribution of viral load to evaluate the role of super spreaders or the changing community infection prevalence that would impact daily exogenous infection rates. Future sensitivity analysis could also evaluate our assumptions around testing compliance, as students at colleges are known to have not received mandated regular testing and to have continued to participate in campus activities for substantial periods of time despite receiving a positive test result.

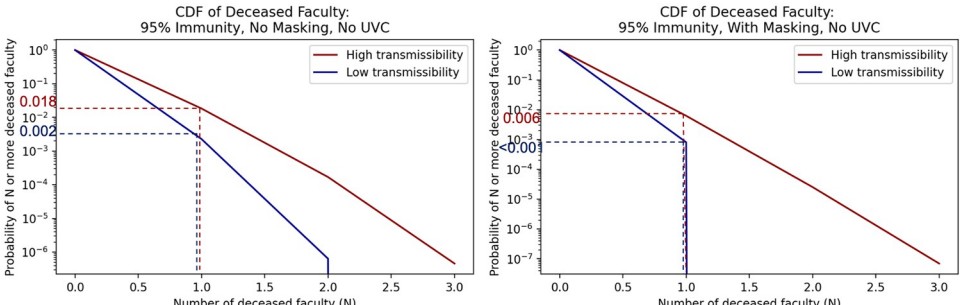

**Fig 9. Inverse cumulative density functions for the total number of faculty that die under the different scenarios.**
Each graph shows the curve of the probability of exceeding a given number of deaths.

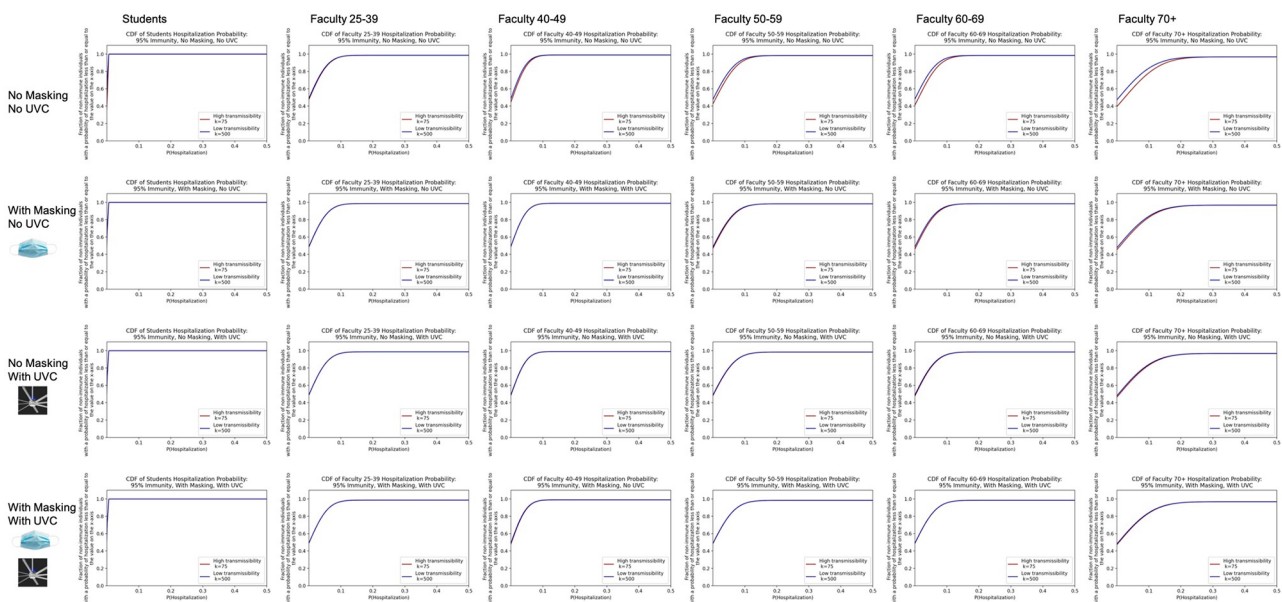

**Fig 10. Cumulative density function for the probability of hospitalization by age group given 95% immunity.** Note that all students are lumped together into one age group, assumed to be the 18–24 age group.

## Discussion

These results highlight how this method can support risk-informed cost-benefit analysis of intervention policies and prioritization of selected mitigation policies. Although UVC fans are potentially more effective than a mask mandate and the impact of UVC fans does not rely on the compliance of individual behavior as is the case with mask use, countervailing factors such

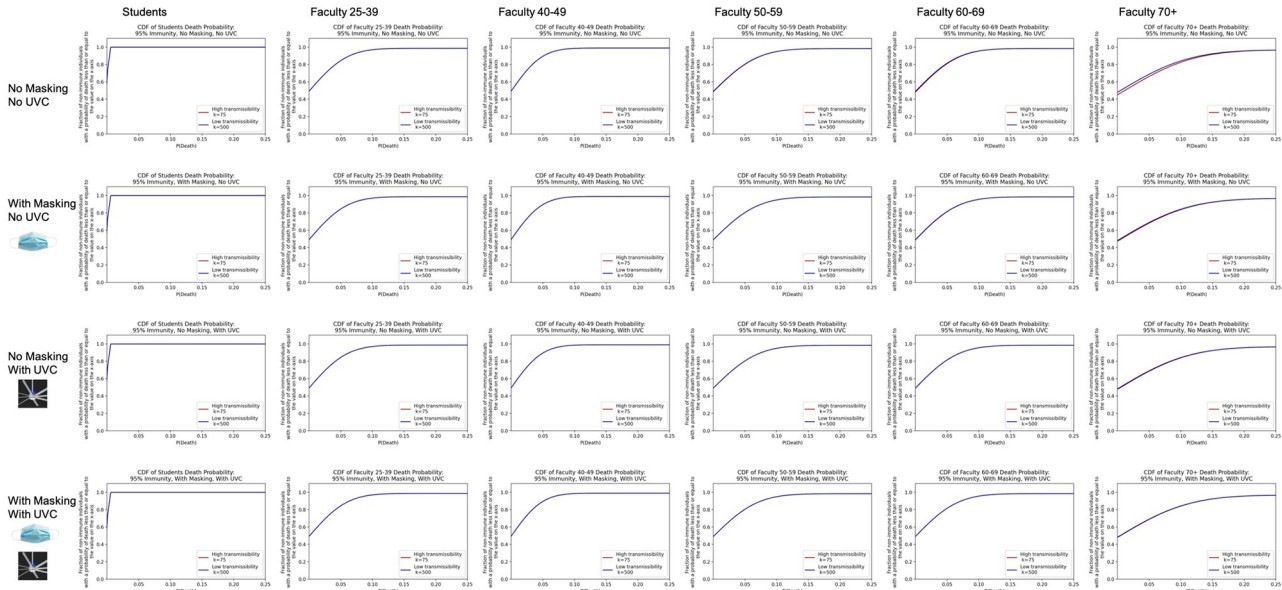

**Fig 11. Cumulative density function for the probability of death by age group given 95% immunity.** Note that all students are lumped together into one age group, assumed to be the 18–24 age group.

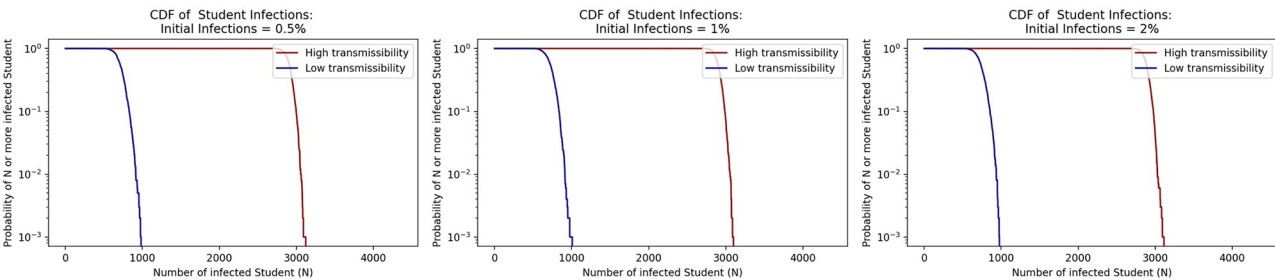

**Fig 12. Sensitivity of model to initial infections rate parameter.**

as cost, time required for installation, and other facility level limitations may prohibit the installation of UVC fans in every classroom. There does remain uncertainty about the rate of deactivation of virions from UVC fans, and this likely depends on specific configurations and rooms and requires careful engineering design. This simulation method could be used to show where fan installation would have the biggest impact on risk and account for relevant opportunity costs to optimize the use of such equipment, even quantifying how many UVC fans are required to meet the same expected risk as masking policies thus quantifying the cost of lifting a mask mandate without increasing risk. Of particular note, masking and UVC effectiveness are relatively insensitive to new variants or other novel aerosol viruses, unlike vaccines whose effectiveness is primarily restricted to the particular vectors for which they were developed.

In high uncertainty scenarios, evaluating such interventions and policies must reflect a range of possibilities for variables outside of the decision makers' control. The variability in vaccine adoption and effectiveness across geographies, ideologies, age groups, and virus variants in the U.S. impacts the effectiveness of the interventions over which the college has control. Population level immunity rates change the magnitude of the impacts of masking and UVC interventions, thereby affecting the cost-benefit ratio of these interventions. Including these exogenous variables in risk-based simulation modeling yields results that can inform decision making and policy prioritization under low-data regimes and remain helpful as the situation and data progress.

Some will likely also ask why in-room HEPA filters were not included as a mitigation option in our demonstration. We initially did include in-room filtration in the model. HEPA filters reduce virus concentrations in the air through filtration. In conversations with facility and instructional planners at our example college there were clear concerns about the noise of HEPA filters used at a high enough setting to be effective for virus removal. In the case of our example college nearly all classrooms are used for recording lectures, making noise a

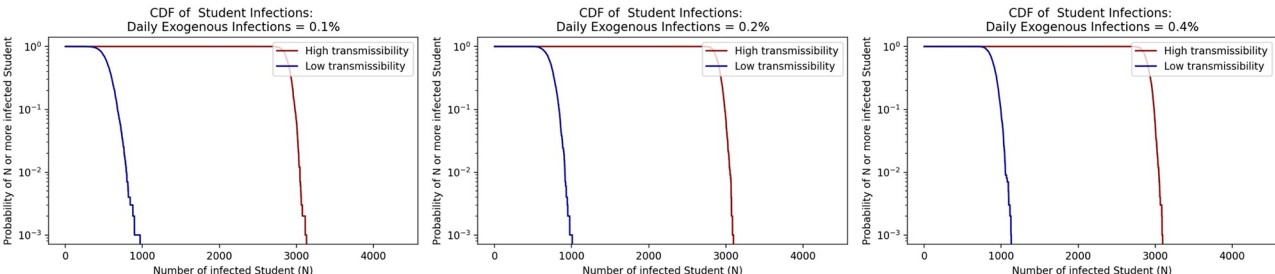

**Fig 13. Sensitivity of model to daily community prevalence (e.g., exogenous infections rate parameter in the model).**

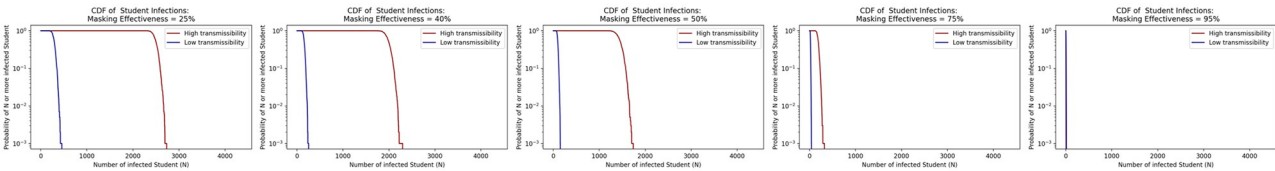

**Fig 14. Sensitivity of model to mask effectiveness parameter.**

substantial concern. In other settings or if lower-noise HEPA filters were available these could be a viable option to consider and can be modeled.

In addition, it should be noted that vaccination is, by itself, a fragile policy. If a new variant emerges that has a higher degree of escape of the vaccine and other mitigation interventions are not in place, the college faces a substantial likelihood of increased transmission and health impacts as well an increased chance of needing to return to virtual instruction. Defense in Depth, the use of multiple layers of interventions, is a critical risk analysis approach to consider in cases of high uncertainty about future conditions in order to have a response that is more robust and resilient in the face of new and varying threats. A Defense in Depth approach means there are still at least some protections (e.g., UVC fans, masking) in place if one layer of protection (e.g., vaccines) fails or is dramatically reduced in its effectiveness. This provides a level of fault tolerance.

## Limitations

While risk-based simulation offers the opportunity to model many scenarios related to aerosol spread in classrooms, these models are still limited by time, data, and the uncertainties of the reality being modeled. Computational run time and data availability force model and parameter assumptions such that no model can perfectly represent a real-world scenario. In this case, adopting the assumption of a well-mixed room over computing the fluid dynamics of each classroom ignores the role of airflow patterns and proximity to the infectious source on transmission. These can be critical in some rooms. Moreover, a model can only be as good as its input data, so any inaccuracy in scheduling or room data impact how representative results can be of the future. For example, if airflow rates in classrooms are significantly lower than those we used in this model, the results may understate risks associated with a do-nothing scenario and the potential benefits of facility interventions. In addition to the inaccuracies of data we do have access to, the lack of epidemiological data on COVID-19 (including variant transmissibility, vaccine efficacy against variants, and dose response mechanisms) limit the accuracy of these results. As such, we consider the role of this model in portraying relative risk across possible scenarios, rather than claiming to precisely predict outcomes. Finally, this simulation can only capture some dimensions of the risk involved with in-person instruction, namely the risk of infection, hospitalization, and death. Risks associated with the economic, reputational, social, and educational costs of virtual learning for faculty and students should also factor into decisions regarding in-person operations. Also, the contributions of other risks outside of in-person classroom activity need to be considered when arriving at a decision of a global systems-based nature.

## Conclusion

The simulation method for evaluating risk associated with aerosol spread of SARS-CoV-2 in classes at a college developed in this paper contributes valuable information to decision makers

during a time of crisis as decisions around that crisis evolve. These results show how simulation modeling can provide valuable data for risk analysis and ultimately decision making even under great uncertainty. Particularly, we show how decisions based on relative risk and the effectiveness of interventions and combinations of interventions to provide defense in depth can lead to a more robust and resilient prioritization of policies and actions. Decision makers can have risk and cost analysis on hand to manage risk levels before information such as student body immunity rates or variant transmissibility parameters are available. Decision makers are then further empowered to adapt those analyses as more data becomes accessible over time.

This simulation model has the capability of expanding into modeling relative risks of additional interventions in response to COVID-19, such as universal testing policies or density protocols. Furthermore, this model has the potential for adoption beyond the example college including to other colleges and universities as well as high-risk population centers such as K-12 schools, day care facilities, or nursing homes. Beyond SARS-CoV-2, this model can model aerosol spread of other diseases ranging from annual influenza strains to future pandemics.

Despite this capacity for expansion into other populations and transmission scenarios, this simulation model on its own cannot implement change. Responsible risk communication that includes not only the decisions and actions taken but also the accompanying rationale is an essential component of risk analysis for translating model results into actual meaning for stakeholders. Appropriate risk communication involves tailoring the reporting of model results to different audiences. For decision makers like college leadership, risk communication must account for highlighting and explaining information in a way that enables individuals to make decisions with that information. For other stakeholders such as faculty and students, risk communication must account for the value of transparency for informed consent, holding leadership accountable for their decisions, and the communication of the residual risk that was accepted.

Beyond the COVID-19 pandemic, universities must be able to cope with many potential and often ignored hazards that affect in-person operations to keep their students, faculty, staff, and surrounding community members acceptably safe. This proactive identification of hazards and their associated risk is an essential component for any organization to avoid the oft heard 'unanticipated' outcomes that many organizations identify as the cause of many tragedies but are really the result of a lack of proactive imagination and effort. Simulation-based risk analysis is a critical tool that helps decision-makers prepare for, mitigate against, adapt to, and recover from such events to evaluate the impacts of variables within and outside of decision-makers control. We should continue developing simulation models for risk analysis to expand our toolbox to better plan for and communicate risk to foster safer and more resilient communities.

## Supporting information

**S1 Appendix. Algorithm structure.**
(ZIP)

**S2 Appendix. Model equations.**
(ZIP)

**S3 Appendix. Inverse cumulative density functions for various outcomes in the college under the different scenarios.** Each graph shows the curve of the probability of exceeding a given number of infections, hospitalizations, or deaths.
(ZIP)

**S4 Appendix. Individual probability of outcomes by immunity rates and age group.**
(ZIP)

# Acknowledgments

We thank the anonymous college for providing the data for this work. We also thank Nancy Love, Jon Zelner, and Charles Haas for providing invaluable input and early reviews of this paper. We thank Advanced Research Computing at our institution, the University of Michigan, for providing time on their compute cluster for some of the model runs used in this paper. While the help and support of all of these individuals and groups are gratefully acknowledged, the results and opinions in the paper are those of authors alone and do not necessarily represent the views of our sponsor or our institution. This study was reviewed an approved by the University of Michigan Institutional Review Board as study number HUM00206822.

# Author Contributions

**Conceptualization:** Tessa Swanson, Seth Guikema, James Bagian.

**Data curation:** Tessa Swanson, Christopher Schemanske.

**Formal analysis:** Tessa Swanson, Christopher Schemanske.

**Funding acquisition:** Tessa Swanson, Seth Guikema, James Bagian.

**Investigation:** Tessa Swanson, Seth Guikema.

**Methodology:** Tessa Swanson, Seth Guikema.

**Project administration:** Seth Guikema, James Bagian.

**Resources:** Seth Guikema, James Bagian.

**Software:** Tessa Swanson.

**Supervision:** Seth Guikema, James Bagian.

**Validation:** Christopher Schemanske.

**Visualization:** Tessa Swanson, Claire Payne.

**Writing – original draft:** Tessa Swanson, Seth Guikema, Christopher Schemanske.

**Writing – review & editing:** Tessa Swanson, Seth Guikema, James Bagian, Claire Payne.

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
