## [Decision Letter · Decision Letter 0]

28 Feb 2022

PONE-D-21-37421COVID-19 aerosol transmission simulation-based risk analysis for in-person learningPLOS ONE

Dear Dr. Swanson,

Thank you for submitting your manuscript to PLOS ONE. After careful consideration, we feel that it has merit but does not fully meet PLOS ONE’s publication criteria as it currently stands. Therefore, we invite you to submit a revised version of the manuscript that addresses the points raised during the review process. In particular, please address the issue around the lack of feedback between infection risk and the prevalence of infection from exogenous transmission.

We look forward to receiving your revised manuscript.

Kind regards,

Nicky McCreesh

Academic Editor

PLOS ONE

Journal Requirements:

2. Please provide additional details regarding participant consent. In the Methods section, please ensure that you have specified (1) whether consent was informed and (2) what type you obtained (for instance, written or verbal). If your study included minors, state whether you obtained consent from parents or guardians. If the need for consent was waived by the ethics committee, please include this information

“This work is funded by the University of Michigan College of Engineering's COVID-19 Skunkworks project.”

Reviewers' comments:

Reviewer's Responses to Questions

**Comments to the Author**

1. Is the manuscript technically sound, and do the data support the conclusions?

Reviewer #1: Partly

Reviewer #2: Yes

2. Has the statistical analysis been performed appropriately and rigorously? 

Reviewer #1: I Don't Know

Reviewer #2: N/A

3. Have the authors made all data underlying the findings in their manuscript fully available?

Reviewer #1: No

Reviewer #2: Yes

4. Is the manuscript presented in an intelligible fashion and written in standard English?

Reviewer #1: Yes

Reviewer #2: Yes

5. Review Comments to the Author

Reviewer #1: The authors are to be commended in developing a risk based model to look at the impact of more than one environmental intervention. Often in risk mitigation there is an administrative component to address impacts of crowding on airborne pathogens, e.g., tuberculosis and crowded outpatient clinics where an undiagnosed case may intermix with patients on treatment. This model looked at one environmental intervention, UVC and one administrative, medical masking of faculty and patients. Overall, the impact of immunity, masking and UVC were shown additive. A few style aspects were difficult where sentences were started with a reference number. Usually author names are listed followed by the reference, for example line 133. This should be corrected throughout the paper. The sentence starting line 152 Apparent across... and ending line 155, doesn't make sense and needs to be re-written. The reference [3] Evans in not peer reviewed, yet is used a number of times in the this paper. It is unclear how this should be handled but it does add a further uncertainty if an eventual peer review challenges some of its assumptions. The UVC fan data related to your assumptions should have a citation or link to review the full report. It was difficult for this reviewer to assess this aspect fully. For example it is known that UVC readily inactivates SARS-CoV-2 on surfaces; however, inactivation data on bioaerosols of SARS-CoV-2 are not yet published in peer review literature. Hopefully, modeling such as this could look at pre and post application of interventions to help determine lessening of risk.

Reviewer #2: The authors analyze the chance of infection (and the risks associated with) transmission of covid-19 in classes. Overall, it is fine study, and it belongs to a large literature being developed around simulation of transmission of COVID-19 in universities. They develop a simple model and provide several estimations.

The results of this paper are generally trivial, mainly suggesting the benefits of using mask and fans in classrooms. So the main contribution is about introducing and reporting a model. The article can be also looked as offering a proof-concept analysis, and an example of how to estimate the risks and COVID-19 transmission in university classes. Even though the contributions are limited and findings are largely expected, in my opinion, it meets the 7 requirements that PLOS ONE is looking for in submissions, thus it is publishable. I have one major questions and a few minor points:

1. Major question:

I might be wrong, but it appeared to me that prevalence is exogenous to the model. One would expect that our policies in classrooms will affect the number of cases, and the future prevalence in the university. That is the fundamental feature of SEIR models (the infection feedback loop), and if you are assuming exogenous prevalence, that would be a major assumption. Given that you have done all the analysis I am reluctant to ask you to endogenize prevalence, thus at least the assumption should be stated clearly and the implications in your findings should be discussed. Note that, if prevalence is assumed exogenous, then you are underestimating the effects of masks and air conditioning. If this is my mis-understanding, and you are consistent with the basic of SEIR, then your figure 1 should show how conceptually new cases, before diagnosis, will lead to more infection in class (can be done by a minor edit).

2. Minor comments:

2.1. It appears that before submission you changed your citation style, and in several places now you need to re-write your phrases. For example see your sentences in page 6 (e.g., it says: [20] use agent-based modeling ....; or you say: [20] and [22] use ....)

2.2. Table 1 did not make sense to me if these numbers of probabilities given infection. The probability of a 70+ dying after infection is not 36%! Did you mean 3.6%? Or maybe you mean 36% chance of death given hospitalization? If so, clarify. The same issue for several other values in the table.

2.3. Page 12: I did not find Appendix 6 (mentioned in two places). Did you mean S1?

2.4 Some of the tables can be presented much better. Tables 4 & 5 for example is too small to read, and Table 3 can turn to a sentence or so. The Figures that were added to the end of the article were hard to read but I noted that your figures in Supplementary were very high quality. This is of course your paper, and feel free to present your results in a way that you are more comfortable with as long as they are readable.

Thank you for the opportunity to read your paper.

6. PLOS authors have the option to publish the peer review history of their article (what does this mean?). If published, this will include your full peer review and any attached files.

Reviewer #1: No

Reviewer #2: No

---

## [Author Response · Author response to Decision Letter 0]

4 May 2022

To our reviewers,

Thank you so much for your time in thoughtfulness in reviewing our manuscript “COVID-19 aerosol transmission simulation-based risk analysis for in-person learning.” We hope you find that your feedback has contributed to a more descriptive and useful paper. Below are responses to each individual reviewer comment:

Reviewer #1: The authors are to be commended in developing a risk-based model to look at the impact of more than one environmental intervention. Often in risk mitigation there is an administrative component to address impacts of crowding on airborne pathogens, e.g., tuberculosis and crowded outpatient clinics where an undiagnosed case may intermix with patients on treatment. This model looked at one environmental intervention, UVC and one administrative, medical masking of faculty and patients. Overall, the impact of immunity, masking and UVC were shown additive. 

A few style aspects were difficult where sentences were started with a reference number. Usually author names are listed followed by the reference, for example line 133. This should be corrected throughout the paper. 

Thank you for bringing our attention to this discrepancy—those sentences previously starting with a reference have been amended to name the authors preceding the reference number.

The sentence starting line 152 Apparent across... and ending line 155, doesn't make sense and needs to be re-written. 

Sentence edited to say: “Each of these fellow researchers’ simulation model results consistently show the effectiveness of defense in depth, where the compounding effects of multiple interventions are necessary for maintaining manageable levels of transmission risk.”

The reference [3] Evans in not peer reviewed, yet is used a number of times in this paper. It is unclear how this should be handled but it does add a further uncertainty if an eventual peer review challenges some of its assumptions. 

• We added additional references for explicitly peer reviewed values including:

o Bazant MZ, Bush JW. A guideline to limit indoor airborne transmission of COVID-19. Proceedings of the National Academy of Sciences. 2021;118(17). 

o Hallett S, Toro F, Ashurst JV. Physiology, Tidal Volume. In: StatPearls. Treasure Island (FL): StatPearls Publishing; 2021. 

• We use this equation from Evans which is not yet peer reviewed, but clarify other formulations may be more appropriate in future application of this model

• We added statement in “General Approach”: “Note, at the time of publication this formulation from (Evans, 2020) was not yet peer reviewed, but the equation utilized for calculating cumulative exposure is verifiable by unit calculations and could be replaced in future applications of the model if appropriate.”

The UVC fan data related to your assumptions should have a citation or link to review the full report. It was difficult for this reviewer to assess this aspect fully. For example it is known that UVC readily inactivates SARS-CoV-2 on surfaces; however, inactivation data on bioaerosols of SARS-CoV-2 are not yet published in peer review literature.

• We added references to manuscript and Supplementary Materials (S1 File) both referencing the inactivation of airborne viruses in general (including coronaviruses) and SARS-CoV-2 on surfaces:

o Ko G, First MW, Burge HA. The characterization of upper-room ultraviolet germicidal irradiation in inactivating airborne microorganisms. Environmental

health perspectives. 2002;110(1):95–101.

o Kowalski W, Bahnfleth WP, Witham D, Severin B, Whittam T. Mathematical modeling of ultraviolet germicidal irradiation for air disinfection. Quantitative microbiology. 2000;2(3):249–270.

o Kowalski, W. J. (2009). Ultraviolet Germicidal Irradiation Handbook: UVGI for Air and Surface Disinfection. Springer, New York.

o Ultraviolet air and surface treatment. In: 2019 ASHRAE Handbook – HVAC

Applications; 2019. p. 1–18.

o Stewart, E. J., Schoen, L. J., Mead, K., Olmsted, R. N., Sekhar, C., Vernon, W., ... & Conlan, W. (2020). ASHRAE position document on infectious aerosols. ASHRAE: Atlanta, GA, USA.

o Biasin, M., Bianco, A., Pareschi, G., Cavalleri, A., Cavatorta, C., Fenizia, C., ... & Clerici, M. (2021). UV-C irradiation is highly effective in inactivating SARS-CoV-2 replication. Scientific Reports, 11(1), 1-7.

o Storm, N., McKay, L. G., Downs, S. N., Johnson, R. I., Birru, D., de Samber, M., ... & Griffiths, A. (2020). Rapid and complete inactivation of SARS-CoV-2 by ultraviolet-C irradiation. Scientific Reports, 10(1), 1-5.

o Buonanno, M., Welch, D., Shuryak, I., & Brenner, D. J. (2020). Far-UVC light (222 nm) efficiently and safely inactivates airborne human coronaviruses. Scientific Reports, 10(1), 1-8.

o Beggs, C. B., & Avital, E. J. (2020). Upper-room ultraviolet air disinfection might help to reduce COVID-19 transmission in buildings: a feasibility study. PeerJ, 8, e10196.

o BIG ASS FANS ION TECHNOLOGY; 2020. https://cleanairsystem.com/technology/?section=uv-c-technology

• Edited sentence in manuscript for clarification: 

“To determine the effect of UVC fans, we calculate an equivalent viral decay rate based on parameters provided by the UVC fan manufacturers (S1 File) consistent with literature on inactivation models for germicidal UVC (Ko 2002; Kowalski 2000). This is not meant to be the definitive assessment of the decay parameters for any UVC intervention, particularly since precise inactivation data on aerosols of SARS-CoV-2 are not yet published in peer reviewed literature. Rather, these parameters give us a reasonable, even conservative, starting point for our demonstration of structural environmental interventions and are based on experiments with live SARS-CoV-2 virus on surfaces (Biasin 2021; Storm 2020), other coronaviruses as aerosols (Buonanno 2020; Kowalski 2009), and recommendations from the American Society of Heating, Refrigeration, Air-Conditioning Engineers (Ultraviolet air and surface treatment 2019; Stewart 2020).”

• Edited description S1 File: 

“The effect of UVC on virion deactivation were based on data provided by the manufacturer of a particular type of fan-mounted, up-shining UVC, Big Ass Fans (Big Ass Fans 2020; Crist 2020). This data was based on chamber simulations experiments using irradiation susceptibility constants (k = 0.377 m2/J) for live SARS-CoV-1 from (Kowalski 2010) and inactivation modeling in Equation (1) of (Kowalski 2000) in line with design recommendations from the American Society of Heating, Refrigeration, Air-Conditioning Engineers (Ultraviolet air and surface treatment 2019; Stewart et al. 2020). This k-value from SARS CoV-1 is considered conservative (note the larger the k-value, the more susceptible a pathogen is to UVC), as evidence mounts that this value is greater than those found for SARS CoV-2 in liquid and on surfaces (Biasin et al. 2021; Storm et al. 2020), and SARS-CoV-2 is demonstrated to be an order of magnitude more susceptible to UVC deactivation when aerosolized (Beggs & Avital 2020).” 

Hopefully, modeling such as this could look at pre and post application of interventions to help determine lessening of risk. 

A primary contribution of this paper is a method for assessing relative risk associated with various interventions, so it is difficult to compare the post application of all interventions. However, we are working on future work validating the scenario that is implemented to identify differences between actual and simulated numbers of infections, hospitalizations, and deaths.

Reviewer #2: The authors analyze the chance of infection (and the risks associated with) transmission of covid-19 in classes. Overall, it is fine study, and it belongs to a large literature being developed around simulation of transmission of COVID-19 in universities. They develop a simple model and provide several estimations.

The results of this paper are generally trivial, mainly suggesting the benefits of using mask and fans in classrooms. So the main contribution is about introducing and reporting a model. The article can be also looked as offering a proof-concept analysis, and an example of how to estimate the risks and COVID-19 transmission in university classes. Even though the contributions are limited and findings are largely expected, in my opinion, it meets the 7 requirements that PLOS ONE is looking for in submissions, thus it is publishable. I have one major question and a few minor points:

1. Major question:

I might be wrong, but it appeared to me that prevalence is exogenous to the model. One would expect that our policies in classrooms will affect the number of cases, and the future prevalence in the university. That is the fundamental feature of SEIR models (the infection feedback loop), and if you are assuming exogenous prevalence, that would be a major assumption. Given that you have done all the analysis I am reluctant to ask you to endogenize prevalence, thus at least the assumption should be stated clearly and the implications in your findings should be discussed. Note that, if prevalence is assumed exogenous, then you are underestimating the effects of masks and air conditioning. If this is my mis-understanding, and you are consistent with the basic of SEIR, then your figure 1 should show how conceptually new cases, before diagnosis, will lead to more infection in class (can be done by a minor edit).

Figure 1 has been updated to show how in-class cases remain in the model until they are removed either via showing symptoms or testing positively. To reiterate, the model does include endogenous cases that occur in the classroom in the following ways:

• Symptomatic individuals are infectious in class for 1 day before their symptoms prevent them from coming to class

• Asymptomatic individuals are infectious in class up until 1 day after they are next tested. So, if an individual is regularly tested on Mondays and is infected on a Tuesday, they become infectious in all of their classes starting Thursday (to account for a 2-day lag period) and remain infectious in all of their classes until the following Tuesday.

2. Minor comments:

2.1. It appears that before submission you changed your citation style, and in several places now you need to re-write your phrases. For example see your sentences in page 6 (e.g., it says: [20] use agent-based modeling ....; or you say: [20] and [22] use ....)

Thank you for bringing our attention to this discrepancy—those sentences previously starting with a reference have been amended to name the authors preceding the reference number.

2.2. Table 1 did not make sense to me if these numbers of probabilities given infection. The probability of a 70+ dying after infection is not 36%! Did you mean 3.6%? Or maybe you mean 36% chance of death given hospitalization? If so, clarify. The same issue for several other values in the table.

The reviewer correctly inferred that this value refers the mean conditional probability, so the probability of death given hospitalization. We clarify this in the manuscript with the following revisions:

• Figure 1 updated to differentiate between replications for hospitalizations given infections and deaths given hospitalizations

• Table 1 updated to clarify hospitalization values are given infection and death values are given hospitalization

• Line 303: “At the end of the semester, we simulate two possible adverse health outcomes of infection: hospitalization given infection and death given hospitalization.”

2.3. Page 12: I did not find Appendix 6 (mentioned in two places). Did you mean S1?

Appendix references have been updated to refer to the corresponding supplementary material.

2.4 Some of the tables can be presented much better. Tables 4 & 5 for example is too small to read, and Table 3 can turn to a sentence or so. The Figures that were added to the end of the article were hard to read but I noted that your figures in Supplementary were very high quality. This is of course your paper, and feel free to present your results in a way that you are more comfortable with as long as they are readable.

• Table 3 is converted to the sentence: “Ultimately, 20 scenarios are evaluated across five different immunity rates (60%, 70%, 80%, 90%, 95%), with and without masking, and with and without UVC fans.”

• Tables 3-6 (formerly 4-7) have been expanded to take up the rest of the page width

• All manuscript figures have been reformatted for readability and compliance for submission-- in the author preview they still appear to be too low resolution, but upon download the quality matches those in the supplementary materials.

Finally, we have reviewed all previous references to ensure no retractions. We did not remove any references but we have included the following additional references throughout the manuscript, primarily in response to reviewer comments and also described above.

• Bazant MZ, Bush JW. A guideline to limit indoor airborne transmission of COVID-19. Proceedings of the National Academy of Sciences. 2021;118(17). 

• Goyal R, Hotchkiss J, Schooley RT, De Gruttola V, Martin NK, et al. Evaluation of SARS-CoV-2 transmission mitigation strategies on a university campus using an agent-based network model. Clinical Infectious Diseases. 2021.

• Hallett S, Toro F, Ashurst JV. Physiology, Tidal Volume. In: StatPearls. Treasure Island (FL): StatPearls Publishing; 2021. 

• Ko G, First MW, Burge HA. The characterization of upper-room ultraviolet germicidal irradiation in inactivating airborne microorganisms. Environmental

health perspectives. 2002;110(1):95–101.

• Kowalski W, Bahnfleth WP, Witham D, Severin B, Whittam T. Mathematical modeling of ultraviolet germicidal irradiation for air disinfection. Quantitative microbiology. 2000;2(3):249–270.

• Biasin M, Bianco A, Pareschi G, Cavalleri A, Cavatorta C, Fenizia C, et al. UV-C irradiation is highly effective in inactivating SARS-CoV-2 replication. Scientific Reports. 2021;11(1):1–7.

• Storm N, McKay LG, Downs SN, Johnson RI, Birru D, de Samber M, et al. Rapid and complete inactivation of SARS-CoV-2 by ultraviolet-C irradiation. Scientific Reports. 2020;10(1):1–5.

• Buonanno M, Welch D, Shuryak I, Brenner DJ. Far-UVC light (222 nm) efficiently and safely inactivates airborne human coronaviruses. Scientific Reports. 2020;10(1):1–8.

• Kowalski W. Ultraviolet germicidal irradiation handbook: UVGI for air and surface disinfection. Springer science & business media; 2010.

• Ultraviolet air and surface treatment. In: 2019 ASHRAE Handbook – HVAC Applications; 2019. p. 1–18.

• Stewart EJ, Schoen LJ, Mead K, Olmsted RN, Sekhar C, Vernon W, et al. ASHRAE position document on infectious aerosols. ASHRAE: Atlanta, GA, USA. 2020.

• Beggs CB, Avital EJ. Upper-room ultraviolet air disinfection might help to reduce COVID-19 transmission in buildings: a feasibility study. PeerJ. 2020;8:e10196.

• BIG ASS FANS ION TECHNOLOGY; 2020. https://cleanairsystem.com/technology/?section=uv-c-technology.

• Crist R. Can this smart ceiling fan kill the coronavirus? Independent tests say ’yes’; 2020. https://www.cnet.com/home/smart-home/can-this-big-ass-fans-haiku-uvc-smart-ceiling-fan-kill-covid-19-independent-tests-say-yes-coronavirus/

 

Thank you again for your time and attention in our efforts to publish this work.

Sincerely,

Tessa Swanson (corresponding author)

Ph.D. Candidate

Department of Industrial and Operations Engineering

University of Michigan

tlswan@umich.edu

Seth Guikema

Professor

Department of Industrial and Operations Engineering

University of Michigan

sguikema@umich.edu

James Bagian

Professor

Department of Industrial and Operations Engineering

Department of Anesthesiology

University of Michigan

jbagian@med.umich.edu

Christopher Schemanske

Department of Industrial and Operations Engineering

University of Michigan

Claire Payne

Department of Industrial and Operations Engineering

University of Michigan

---

## [Decision Letter · Decision Letter 1]

20 May 2022

PONE-D-21-37421R1COVID-19 aerosol transmission simulation-based risk analysis for in-person learningPLOS ONE

Dear Dr. Swanson,

Thank you for submitting your manuscript to PLOS ONE. After careful consideration, we feel that it has merit but does not fully meet PLOS ONE’s publication criteria as it currently stands. Therefore, we invite you to submit a revised version of the manuscript that addresses the points raised during the review process.

 I would like to give you the opportunity to make any additional changes you wish in response to the reviewer's comments, before the paper is accepted. Please ensure that your decision is justified on PLOS ONE’s publication criteria and not, for example, on novelty or perceived impact.

We look forward to receiving your revised manuscript.

Kind regards,

Nicky McCreesh

Academic Editor

PLOS ONE

Journal Requirements:

Reviewers' comments:

Reviewer's Responses to Questions

**Comments to the Author**

1. If the authors have adequately addressed your comments raised in a previous round of review and you feel that this manuscript is now acceptable for publication, you may indicate that here to bypass the “Comments to the Author” section, enter your conflict of interest statement in the “Confidential to Editor” section, and submit your "Accept" recommendation.

Reviewer #2: All comments have been addressed

2. Is the manuscript technically sound, and do the data support the conclusions?

Reviewer #2: Yes

3. Has the statistical analysis been performed appropriately and rigorously? 

Reviewer #2: N/A

4. Have the authors made all data underlying the findings in their manuscript fully available?

Reviewer #2: Yes

5. Is the manuscript presented in an intelligible fashion and written in standard English?

Reviewer #2: Yes

6. Review Comments to the Author

Reviewer #2: I appreciate the authors' response to my comments, and I am particularly happy that they consider infection endogenously in their model. The background section of the paper can be further improved (still citation issues, and in many places you can re-write to avoid listing names of authors of papers that are only marginally related to your work). Otherwise it looks fine to me.

Just for the authors' information, I would like to write that SEIR is referred to any model that categorizes people in these groups, but that doesn't mean that an agent-based model is not an SEIR. In your writing it felt like you use SEIR to refer to "Compartmental" models (yes, all compartmental models of infection are SEIR-like models), and then you have a different category for agent-based model. A more accurate categorization of SEIR models is compartmental models vs. agent-based models. This is very minor issue and a common mistake in the modeling community. Good luck, and thanks for sharing your paper with me.

7. PLOS authors have the option to publish the peer review history of their article (what does this mean?). If published, this will include your full peer review and any attached files.

Reviewer #2: No

---

## [Author Response · Author response to Decision Letter 1]

28 Jun 2022

We rewrote a portion of our Background section (starting with line 117) to better clarify the differences between agent-based models, compartmental models, and our model presented in this manuscript. We also more concisely summarized the literature corresponding to those models and model differences.

---

## [Editor Report · Decision Letter 2]

7 Jul 2022

COVID-19 aerosol transmission simulation-based risk analysis for in-person learning

PONE-D-21-37421R2

Dear Dr. Swanson,

We’re pleased to inform you that your manuscript has been judged scientifically suitable for publication and will be formally accepted for publication once it meets all outstanding technical requirements.

Kind regards,

Nicky McCreesh

Academic Editor

PLOS ONE
---

## [Editor Report · Acceptance letter]

11 Jul 2022

PONE-D-21-37421R2 

COVID-19 aerosol transmission simulation-based risk analysis for in-person learning 

Dear Dr. Swanson:

I'm pleased to inform you that your manuscript has been deemed suitable for publication in PLOS ONE. Congratulations! Your manuscript is now with our production department. 

Kind regards, 

on behalf of

Dr. Nicky McCreesh 

Academic Editor

PLOS ONE